# Natural changes in light interact with circadian regulation at promoters to control gene expression in cyanobacteria

**Joseph Robert Piechura[1,2,3†], Kapil Amarnath[2,3†], Erin K O'Shea[1,2,3,4*]**

[1]Department of Molecular and Cellular Biology, Harvard University, Cambridge, United States; [2]FAS Center for Systems Biology, Harvard University, Cambridge, United States; [3]Howard Hughes Medical Institute, Harvard University, Cambridge, United States; [4]Department of Chemistry and Chemical Biology, Harvard University, Cambridge, United States

**Abstract** The circadian clock interacts with other regulatory pathways to tune physiology to predictable daily changes and unexpected environmental fluctuations. However, the complexity of circadian clocks in higher organisms has prevented a clear understanding of how natural environmental conditions affect circadian clocks and their physiological outputs. Here, we dissect the interaction between circadian regulation and responses to fluctuating light in the cyanobacterium *Synechococcus elongatus*. We demonstrate that natural changes in light intensity substantially affect the expression of hundreds of circadian-clock-controlled genes, many of which are involved in key steps of metabolism. These changes in expression arise from circadian and light-responsive control of RNA polymerase recruitment to promoters by a network of transcription factors including RpaA and RpaB. Using phenomenological modeling constrained by our data, we reveal simple principles that underlie the small number of stereotyped responses of dusk circadian genes to changes in light.
DOI: https://doi.org/10.7554/eLife.32032.001

**\*For correspondence:**
osheae@hhmi.org

†These authors contributed equally to this work

## Introduction

Circadian clocks allow organisms from almost all branches of life to alter physiology in anticipation of diurnal changes in the environment. Circadian clocks are autonomous core oscillators that keep time even in the absence of environmental cues (*Dunlap et al., 2004*). Output pathways interpret timing information from the core oscillator to generate oscillating outputs, such as oscillations in the mRNA levels (expression) of genes and higher order behaviors (*Dunlap et al., 2004*; *Wijnen and Young, 2006*). Laboratory studies of the outputs of circadian clocks have been primarily performed under constant conditions to isolate circadian regulation from environmental responses. In nature, however, organisms with circadian clocks must also cope with unexpected fluctuations in the environment. Thus a major challenge in chronobiology is to understand circadian regulation in dynamic environments.

Previous studies suggest that circadian clock output pathways interact with responses to the environment to tailor physiology to both the time of day and the current state of the environment. For example, sleep/wake cycles in *Drosophila melanogaster* and photosynthesis in *Arabidopsis thaliana* are controlled by both the circadian clock and environmental variables like day length or light (*Lamaze et al., 2017*; *Millar and Kay, 1996*). Further, circadian clocks can modulate responses to the environment based on the time-of-day in a process called circadian gating (*Hotta et al., 2007*; *Greenham and McClung, 2015*). However, the complexity of higher organisms has prevented a detailed understanding of the interaction between circadian timing information and environmental

**eLife digest** Living things face daily, predictable challenges due to the regular day and night cycle imposed by the Earth's rotation. Many of them have evolved an internal 'circadian' clock to anticipate daily changes in the environment. However, nature can also change in unpredictable ways, and in order to survive, organisms must account for both the time of day stipulated by their clocks and changes in their present environment. For example, cyanobacteria depend on the sun for survival and must cope with light variations throughout the day and the absence of light at nighttime.

Circadian clocks are made up of specific genes and their proteins. Most of what we know about how these clocks control the behavior of an organism comes from experiments performed under constant conditions. Previous research has shown that under such circumstances, the circadian clock of cyanobacteria periodically turns on a set of genes every 24 hours via a protein called RpaA. However, to understand how cyanobacteria use this clock, we must know how it works in a fluctuating environment.

To test this, Piechura, Amarnath and O'Shea measured the activation of genes in cyanobacteria that had been exposed to changes in light mimicking those in nature. Compared to constant conditions, fluctuating light drastically changed the timing of activation of circadian genes. When light decreased – as it would in nature during sunset or if a cloud blocks the sun – the circadian genes were activated.

Changes in light did not change the 'ticking' of the clock, but did affect the ability of RpaA to turn on circadian genes. Moreover, the activity of a second protein called RpaB increased when light decreased and the genes were activated. Thus, cyanobacteria switch on circadian genes as the sun is setting or during unexpected shade, likely through RpaA and RpaB, to help them survive without light.

This study shows that circadian clocks activate genes differently in the real world compared to unnatural, constant conditions. This may prompt scientists to think carefully about how an organism's natural environment can affect its inner workings. A next step will be to see how else light affects circadian gene levels. A deeper understanding of how cyanobacteria control their genes in a natural environment will be useful for scientists who engineer these organisms to produce biofuels from sunlight.

DOI: https://doi.org/10.7554/eLife.32032.002

responses. In contrast, the circadian clock in the cyanobacterium *Synechococcus elongatus* PCC7942, an obligate photoautotroph, has a simple architecture which controls gene expression oscillations (*Figure 1A*) to influence metabolism and growth. *S. elongatus* must carefully monitor its environment, as the sunlight required for photosynthesis fluctuates on the minute, day, and seasonal timescales (*Figure 1B*, [*Petty and Weidner, 2017*]). While it is well understood how the circadian clock in *S. elongatus* behaves under constant conditions, it is unclear how this system changes in natural, fluctuating light.

In *S. elongatus* grown under 'Constant Light' conditions (*Figure 1A*, dashed navy blue line), genes which show oscillatory expression (circadian genes) can be divided into two groups, the dawn and the dusk genes, which peak at subjective dawn and subjective dusk (*Ito et al., 2009*; *Vijayan et al., 2009*) (*Figure 1A*). Subjective dawn and subjective dusk refer to the times at which dark-to-light or light-to-dark transitions would occur in a 12 hr light-12 hr dark environmental cycle. The dawn genes consist of the core metabolic and growth genes for *S. elongatus*, including the photosystems, ATP synthase, carbon fixation/Calvin-Benson-Bassham cycle enzymes, and ribosomal proteins (*Vijayan et al., 2009*; *Ito et al., 2009*; *Diamond et al., 2015*). In the absence of regulation by the circadian clock under Constant Light, *S. elongatus* constantly expresses dawn genes (*Markson et al., 2013*). The clock primarily regulates the expression of dusk genes (*Markson et al., 2013*), which include the genes required to utilize glycogen as an energy source in the absence of sunlight, such as glycogen phosphorylase and cytochrome c oxidase. As such, the circadian clock serves a critical function in switching *S. elongatus* from a daytime state of photosynthesis to a nighttime state of carbon metabolism through glycogen breakdown (*Diamond et al., 2015*;

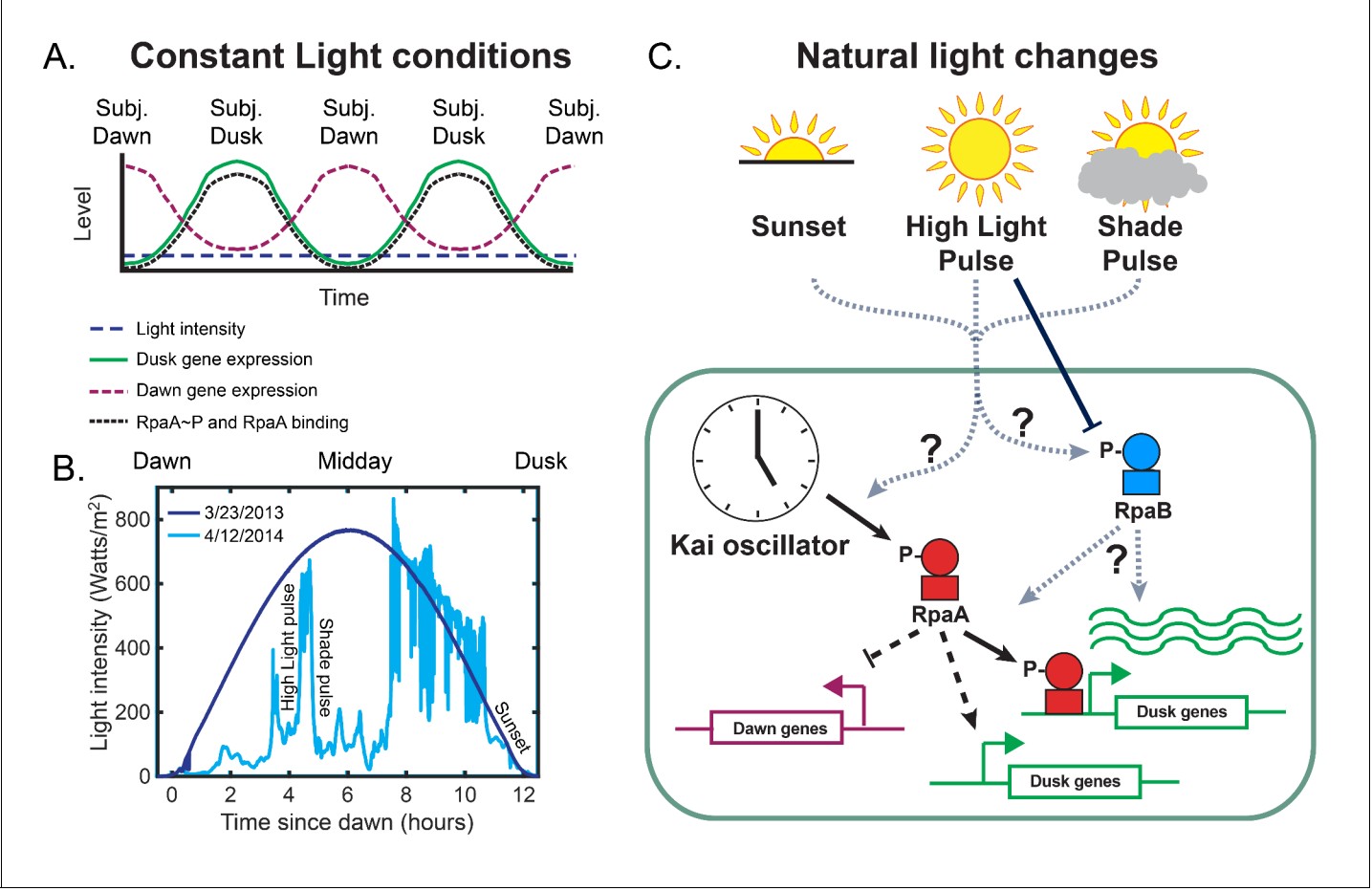

**Figure 1.** The circadian and light response pathways in cyanobacteria. (**A**) Schematic of gene expression output of the circadian clock under Constant Light conditions. Under Constant Light intensity (dashed navy blue line), dawn gene expression (dashed maroon line) and dusk gene expression (solid green line) display oscillatory patterns, peaking at subjective dawn and subjective dusk, respectively. The Kai post-translational oscillator generates oscillations in the levels of phosphorylated RpaA (RpaA~P) and the binding of RpaA to DNA (black dotted line), with the peak amplitude at subjective dusk. (**B**) Solar irradiance measurements in units of watts m$^{-2}$ at 342.5 meters above sea level in Madison, WI, on a clear day (3/23/13, dark blue), and a day on which fluctuations in cloud cover generated rapid changes in light intensity (4/12/14, light blue) (*Petty and Weidner, 2017*). Examples of a 'High Light pulse,' 'Shade pulse,' and 'Sunset' are indicated. (**C**) Schematic of the regulation of circadian gene expression. RpaA phosphorylation state converts timing information from the Kai oscillator to changes in gene expression by directly binding and activating a subset of dusk genes, indirectly activating the remainder of the dusk genes, and indirectly repressing the dawn genes. High Light Pulse conditions cause dephosphorylation of RpaB (*Moronta-Barrios et al., 2012*), but the effects of conditions like Sunset or Shade on RpaB are unknown. It is unclear whether natural fluctuations in light directly affect the clock and its output pathways and how light-induced changes in RpaB activity might be involved.
DOI: https://doi.org/10.7554/eLife.32032.003

*Diamond et al., 2017*; *Pattanayak et al., 2014*; *Puszynska and O'Shea, 2017*). In Constant Light conditions, the dusk and dawn genes show oscillatory expression with a 24 hr period, resulting in broad peaks of maximal expression (*Figure 1A*, solid green line and dashed maroon line) (*Vijayan et al., 2009*; *Ito et al., 2009*). Recent whole-cell modeling of metabolism, protein levels, and growth predict that this picture of circadian gene expression should change under the dynamic light conditions of a natural, clear day (*Figure 1B*, navy blue line) (*Reimers et al., 2017*). The modeling suggests that making and using glycogen is a major cost to cell growth and thus the expression of genes required to switch metabolism from photosynthesis to glycogen breakdown should be delayed until absolutely necessary (*Reimers et al., 2017*). However, gene expression in natural light conditions has not been measured in *S. elongatus*.

Consistent with predictions of light-dependent changes in circadian gene expression, current evidence suggests interaction between the circadian and light regulatory pathways. The cyanobacterial

clock keeps track of the time of day using a core post-translational oscillator (PTO) that consists of three proteins, KaiA, KaiB, and KaiC, whose enzymatic activities result in 24 hr oscillations in the phosphorylation state of KaiC (*Nakajima et al., 2005*; *Rust et al., 2007*; *Johnson et al., 2011*). In vivo under Constant Light conditions the Kai PTO modulates circadian gene expression by controlling oscillations in phosphorylation state of the master OmpR-type transcription factor RpaA (*Markson et al., 2013*; *Takai et al., 2006*) to peak at subjective dusk (*Figure 1A*, dotted black line; *Figure 1C*) (*Gutu and O'Shea, 2013*; *Takai et al., 2006*). Phosphorylated RpaA (RpaA~P) binds to the promoters of some dusk genes to activate their expression, leading to indirect activation of other dusk genes and repression of dawn genes (*Figure 1C*) (*Markson et al., 2013*). As *kaiBC* is a dusk gene target of RpaA, the Kai PTO directs its own expression, resulting in a transcription-translation feedback loop that stabilizes the phase of the clock (*Qin et al., 2010*; *Teng et al., 2013*; *Zwicker et al., 2010*). Exposure to complete darkness at specific times of day causes phase shifts in the PTO to align clock output with the external day/night cycle (*Rust et al., 2011*), in a process called entrainment. However, it is not understood whether any aspect of this model, such as the dynamics of RpaA activity or the transcription-translation feedback loop, changes in the presence of more subtle natural light changes during the day (*Figure 1C*).

Meanwhile the OmpR-type transcription factor RpaB binds to some circadian gene promoters (*Hanaoka et al., 2012*), and the phosphorylation state and DNA binding activity of this protein decreases in response to high light exposure (*Figure 1C*) (*López-Redondo et al., 2010*; *Moronta-Barrios et al., 2012*). However, it is not clear how natural light changes like sunset or shade pulses affect RpaB activity (*Figure 1C*). RpaB clearly plays some role in altering circadian gene expression in response to light (*Espinosa et al., 2015*), but it is unclear how (*Figure 1C*). While light likely exerts global, growth-rate-dependent regulation of protein levels (*Scott et al., 2010*; *Du et al., 2016*; *Burnap, 2015*), the interaction between circadian and light regulation to control the activities of RpaA and RpaB represents a particularly tractable scenario for dissecting the mechanisms underlying interaction between clock and environment to control circadian gene expression.

Here we measure and model circadian gene expression and several layers of regulation in cyanobacteria grown under the fluctuating light intensities typically experienced in nature. We find that fluctuations in light alter the expression patterns of almost all circadian genes. We identify key regulatory steps at which information about changes in light interact with clock output pathways to control gene expression, and reveal a complex regulatory network underlying circadian gene expression in natural conditions. Finally, we show that phenomenological models effectively describe the integration of the circadian clock with responses to environmental fluctuations.

## Results

### Sunlight on a clear day delays the timing of circadian gene expression relative to constant light conditions

To grow and assay cyanobacteria in natural light conditions, we custom-built a culturing setup with a light source that can be programmed to mimic natural fluctuations in sunlight. On a cloudless 'Clear Day,' light intensity varies in a parabolic manner due to the rotation of the Earth, ending with a gradual ramp down of light intensity prior to dusk ('Sunset', *Figure 1B*). Rapid changes in cloud cover cause abrupt increases ('High Light pulse') and decreases ('Shade pulse') in sunlight (*Petty and Weidner, 2017*) (*Figure 1B*). Using a set of programmable warm white LED arrays (Materials and methods, Construction of light apparatus and Calibrating light conditions) for illumination, in all experiments we grew cells for 12 hr in either a Clear Day condition that peaked at 600 $\mu$mol photons m$^{-2}$ s$^{-1}$ or a continuous Low Light condition of 50 $\mu$mol photons m$^{-2}$ s$^{-1}$ (*Figure 2A*, top panel) followed by 12 hr of darkness for at least two days to acclimate and synchronize the cells before measurement. Note that the Low Light condition used here differs from the Constant Light condition (often denoted as LL in the literature; *Figure 1A*, dashed navy blue line) in that the cells are exposed to more naturally-relevant 12 hr light-12 h dark days (LD). Cultures grown under the Clear Day condition adjusted their pigment content after two days of exposure to the Clear Day condition (*Figure 2—figure supplement 1*). Further, cells acclimated to the Clear Day conditions grew approximately twice as fast as Low Light acclimated cultures at midday, 6 hr after dawn (*Figure 2—figure supplement 1*). These data indicate that *S. elongatus* PCC7942 is capable of acclimating to

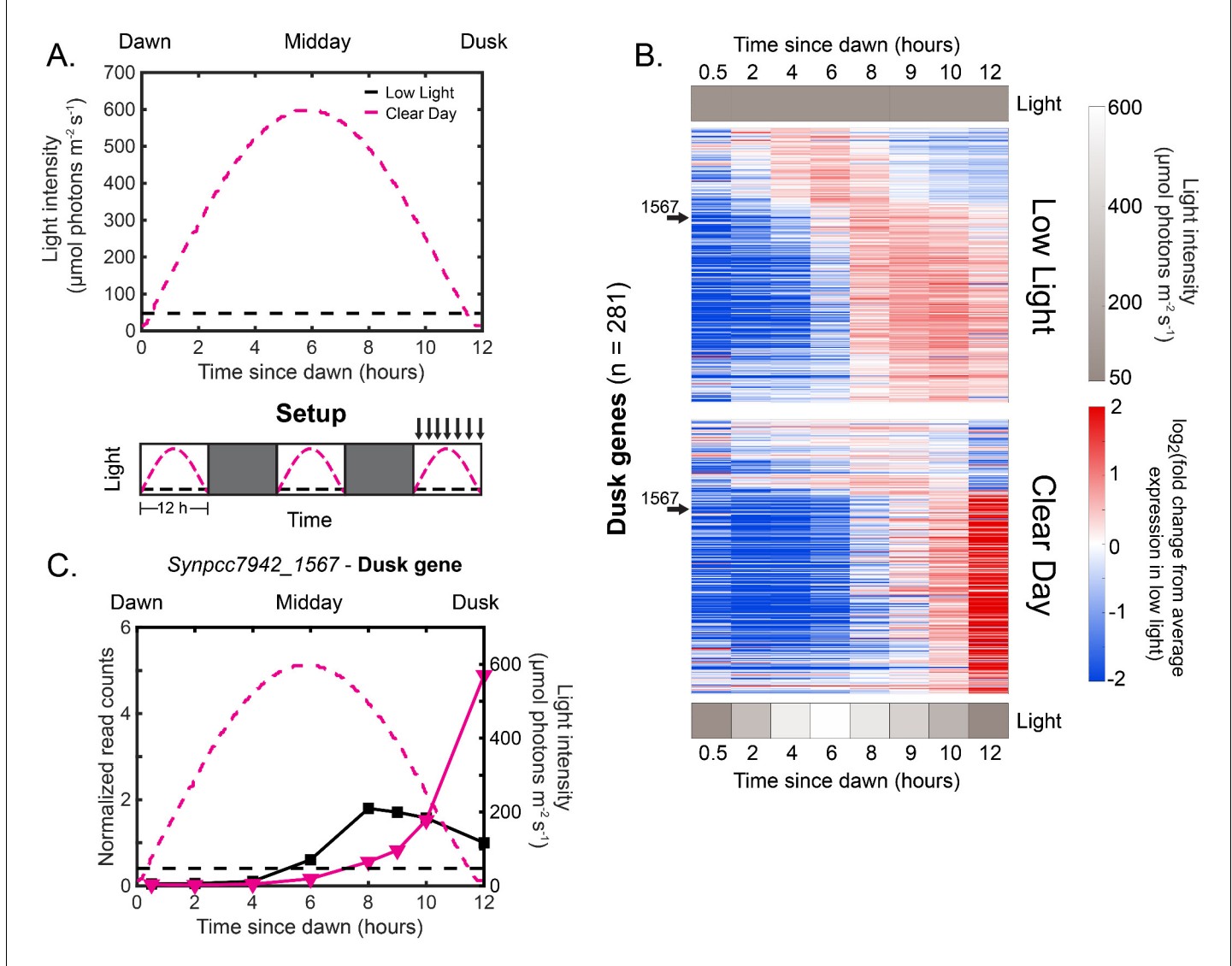

**Figure 2.** Natural clear day conditions sharpen the expression of dusk genes to peak just before expected darkness. (**A**) Experimental setup for testing the effects of Clear Day conditions on circadian gene expression. The upper plot shows the light intensity profiles of Low Light (black) and Clear Day (magenta) conditions, in units of $\mu$mol photons m$^{-2}$ s$^{-1}$ (see Materials and methods - Calibrating light conditions for more details; light intensity values available in *Figure 2—source data 1*). The lower plot displays the experimental setup. Cells were grown under Clear Day (magenta dashed lines) or Low Light conditions (black dashed lines) for 12 hr, followed by 12 hr of darkness (dark gray boxes) for three days, with sampling over the third light period (indicated by arrows above plot). (**B**) Gene expression dynamics of all dusk genes (*n*=281) under Low Light (top) and Clear Day (bottom) conditions. Gene expression is quantified as the log$_2$ fold change from the average expression of the gene over all time points in the Low Light condition (see Materials and methods - RNA sequencing for more details; data available in *Figure 2—source data 1*). Genes were sorted by phase under Constant Light conditions (*Vijayan et al., 2009*). Light intensity at each time point is indicated in a grayscale heat map next to the corresponding condition. The data for a representative dusk gene, *Synpcc7942_1567*, is indicated with arrows. (**C**) Gene expression dynamics of the representative dusk gene *Synpcc7942_1567* under Low Light (black) and Clear Day (magenta) conditions (left y-axis). The light profile for each condition is plotted as dashed lines of the same color with values corresponding to the right y-axis.

DOI: https://doi.org/10.7554/eLife.32032.004

The following source data and figure supplements are available for figure 2:

**Source data 1.** Normalized gene expression in Low Light and Clear Day conditions.

DOI: https://doi.org/10.7554/eLife.32032.009

**Figure supplement 1.** Pigment levels of cyanobacteria grown under Low Light or Clear Day conditions reveal adjustments in the photosynthetic apparatus to optimize growth in different light conditions.

DOI: https://doi.org/10.7554/eLife.32032.005

*Figure 2 continued on next page*

*Figure 2 continued*

**Figure supplement 2.** Gene expression dynamics of dusk and dawn circadian genes under Constant Light conditions (data from *Markson et al., 2013*).

DOI: https://doi.org/10.7554/eLife.32032.006

**Figure supplement 3.** Dawn gene expression increases during the early part of Clear Day relative to Low Light conditions.

DOI: https://doi.org/10.7554/eLife.32032.007

**Figure supplement 4.** The gene expression dynamics of glycogen production and breakdown enzymes change in Clear Day conditions relative to Low Light conditions.

DOI: https://doi.org/10.7554/eLife.32032.008

the higher light intensities of the Clear Day condition and thus that the intensities used in our measurements are relevant for this strain.

To determine whether a natural light profile affects circadian output, we compared genome-wide gene expression in Clear Day conditions versus Low Light conditions using RNA sequencing (*Figure 2A*, Setup, arrows indicate sampling). We acclimated cultures in their respective condition for 2 light/dark cycles, and sampled them (arrows) over the next (third) light period (*Figure 2A*, Setup). We focused our analysis on a set of high amplitude circadian genes that show oscillatory expression under Constant Light conditions (*Figure 2—figure supplement 2*; see Materials and methods, Definition of circadian genes). The Low Light condition (*Figure 2B*, upper panel) reproduces the expression profile previously observed under Constant Light conditions (*Figure 2—figure supplement 2*). However, in the Clear Day condition 159 of the 281 dusk genes were expressed at least two fold higher after midday compared to Low Light, demonstrating light-dependent expression. Dawn genes show the opposite behavior — they have higher expression at midday under Clear Day conditions, although this trend is less pronounced (*Figure 2—figure supplement 3*). Taken together, Clear Day conditions significantly influence the expression dynamics of almost all circadian genes, with the strongest effects on dusk genes.

To look more closely at how the Clear Day condition affects the dusk genes, which are the primary regulatory targets of the clock, we analyze the gene expression dynamics of the representative dusk gene *Synpcc7942_1567*. Under Low Light conditions, *Synpcc7942_1567* exhibits an increase in expression from dawn to dusk, reaching a plateau by 8 hr after dawn (*Figure 2C*, solid black line). Under Clear Day conditions, however, the expression of this gene remains low through the midday peak of light intensity (*Figure 2C*, solid magenta line; 4–8 hr after dawn), and its expression sharply increases just prior to dusk as light intensity decreases, reaching maximal expression just as the dark period begins. This delayed pattern of gene expression can be seen in almost all dusk genes (*Figure 2B*; *Synpcc7942_1567* indicated with arrows). Thus Clear Day conditions significantly alter the dynamics and amplitude of dusk gene expression to peak just before dusk.

The delay of dusk gene expression likely enables cyanobacteria to switch to glycogen breakdown only when absolutely necessary so that they can survive the extended darkness of night. The two glycogen breakdown genes, *glgP* and *glgX*, are both light-dependent dusk genes that strongly peak in Clear Day at dusk, while *glgC*, which codes for the rate limiting enzyme of glycogen synthesis, is a dawn gene whose expression is higher in Clear Day conditions compared to Low Light (*Figure 2—figure supplement 4*). These gene expression dynamics would favor both the maintenance of glycogen synthesis until the end of the day and a delay in the activation of glycogen breakdown until just before it is required at nighttime, in agreement with predictions from metabolic modeling during the same Clear Day conditions used here (*Reimers et al., 2017*). Thus, environmental conditions are integrated into the output of the circadian clock to potentially optimize resource allocation in naturally-relevant diurnal cycles, as recently suggested (*Reimers et al., 2017*).

Remarkably, though in both light conditions the cells experience 50 $\mu$mol photons m$^{-2}$ s$^{-1}$ at the end of the day just before night, light-dependent dusk genes have substantially higher expression in the Clear Day conditions relative to the Low Light conditions (*Figure 2B–C*). Indeed, 95/281 dusk genes were expressed at least three fold higher in Clear Day relative to Low Light at 12 hr after dawn. This strong activation of dusk genes occurs concomitant with the decrease in light intensity during Clear Day that mimics Sunset, which hinted that *changes* in light intensity affect activation of dusk genes as opposed to absolute light intensity levels. Dusk gene expression could thus happen 'just-in-time' before the sustained darkness of nighttime regardless of the seasonal timing of Sunset.

## Changes in light intensity control the transcription of circadian genes

To test whether changes in light intensity are a key factor controlling the expression of circadian genes, we exposed cells to a High Light pulse or a Shade pulse and measured genome-wide gene expression using RNA sequencing. We grew cultures in either Low Light or Clear Day conditions for three days (*Figure 3A–B*, Setup). On the fourth day at 8 hr after dawn, when RpaA is most active, we exposed the cells to a High Light pulse (*Figure 3A*) or a Shade pulse (*Figure 3B*) for 1 hr before returning to the original condition. We sampled the cells before, during, and after the perturbation (*Figure 3A–B*, Setup, arrows). The expression of dusk genes rapidly changed in a direction opposite to the change in light intensity (*Figure 3C*, all dusk genes; *Figure 3E*, example dusk gene; *Figure 3D*, all dusk genes; *Figure 3F*, example dusk gene), as expected from the effects of the decrease in light intensity at Sunset of the Clear Day condition on circadian gene expression (*Figure 2B–C*). A large subset of dusk genes were affected by the light pulses, with 105/281 repressed by at least three fold by the High Light condition, and 136/281 induced by at least three fold by the Shade condition. Further, many genes responded rapidly and changed in expression at least three fold after just 15 min into the pulse (75/281 repressed by High Light, 79/281 induced by Shade). When cultures were restored to their original condition (High Light to Low Light, *Figure 3C, E*; Shade to Clear Day, *Figure 3D,F*), dusk gene expression quickly reverted to a level comparable to that before the pulse. Thus, light-induced changes in dusk gene expression are reversible and responsive to successive shifts in light availability. Dawn gene expression showed the opposite behavior of dusk genes, albeit with less dramatic changes (*Figure 3—figure supplement 1*). Hence, decreases in light intensity favor the expression of dusk genes (Sunset in Clear Day, *Figure 2*; Clear Day to Shade and High Light to Low Light, *Figure 3*), while increases in light favor the expression of dawn genes (midday peak in Clear Day, *Figure 2—figure supplement 3*; Shade to Clear Day and Low Light to High Light, *Figure 3—figure supplement 1*). Given the more substantial effects of light on dusk gene expression, we focus on these genes for the remainder of the manuscript.

To cause these reversible changes in the mRNA levels of dusk genes, changes in light intensity must affect either the transcription and/or the degradation of dusk gene mRNAs. We reasoned that changes in transcription would manifest as differences in the amount of RNA polymerase (RNAP) localized to dusk genes. To determine whether changes in light intensity regulate the recruitment of RNAP to dusk gene promoters, we performed chromatin immunoprecipitation followed by high-throughput sequencing (ChIP-seq) of RNAP in cells immediately before the High Light or Shade pulse (8 hr after dawn in Low Light or Clear Day), and then again 15 or 60 min following the start of the pulse. Changes in RNAP enrichment upstream of dusk genes correlated with changes in down-stream dusk gene expression (*Figure 3G,H*; *Figure 3—figure supplement 2*). Thus, changes in light affect RNAP recruitment to dusk gene promoters, suggesting that light conditions substantially affect the rates of transcription of dusk gene mRNAs. Because mRNAs in bacteria have very short steady state half lives (*Chen et al., 2015*; *Hambraeus et al., 2003*; *Salem and van Waasbergen, 2004*), we argue that changes in transcription rates of dusk gene mRNAs are sufficient to lead to the rapid changes in dusk gene mRNA levels given a fast basal degradation rate, though we cannot rule out that changes in light may affect the rates of degradation of some mRNAs. These results point to a potential interaction between sunlight and signaling pathways upstream of RNAP. We next explored how the observed changes in dusk gene expression in the presence of natural light fluctuations (*Figures 2* and *3*) could be achieved via gene regulatory mechanisms.

## Regulation of dusk gene expression by RpaA and RpaB under dynamic light regimes

Given the strong dependence of dusk gene expression on RpaA~P levels under Constant Light conditions (*Figure 1A*, [*Markson et al., 2013*]) and the drastic change in dusk gene expression dynamics under our dynamic light conditions (*Figure 2B,C*; *Figure 3C–F*), we hypothesized that light conditions alter RpaA~P dynamics to alter dusk gene expression. However, levels of RpaA~P increased from dawn to dusk similarly in cells grown in either Low Light or Clear Day conditions, and abrupt changes in light intensity did not affect these dynamics. (*Figure 4A,B*; *Figure 4—figure supplement 1*). Thus, these natural light fluctuations do not affect the phase of the Kai PTO nor the control of RpaA~P levels by the Kai PTO (*Figure 4E*). These data demonstrate that the regulation of dusk genes is de-coupled from RpaA~P levels under dynamic light conditions, and light must affect

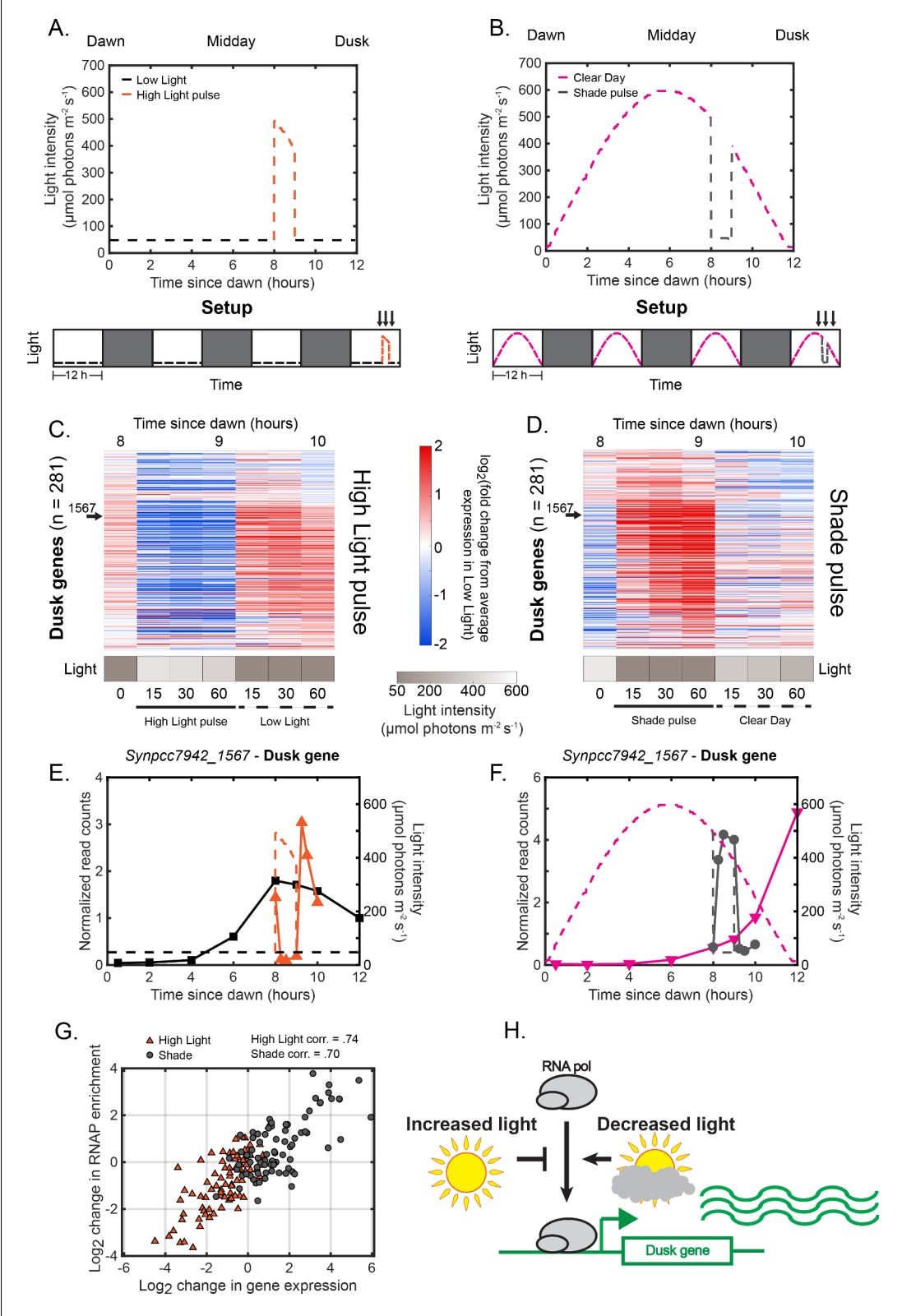

**Figure 3.** Rapid changes in light intensity modulate the recruitment of RNA polymerase to dusk genes to control dusk gene expression. (A) Light intensity profiles of Low Light (black) and High Light pulse (orange) conditions, in units of $\mu$mol photons m$^{-2}$ s$^{-1}$ (see Materials and methods - Calibrating light conditions for more details; light intensity values for pulse conditions available in *Figure 3—source data 1*). Experimental setup is displayed in the lower plot. Cells were grown for 12 hr under Low Light conditions (black dashed lines), followed by 12 hr of darkness (dark gray boxes)

*Figure 3 continued on next page*

Figure 3 continued

for three days, and then exposed to a High Light pulse (dashed orange lines) at 8 hr after dawn during the fourth light period for one hour before being returned to Low Light conditions. Cells were sampled immediately before, during, and after the High Light pulse (indicated by arrows above plot). (Caption continued on next page.). (B) Light intensity profiles of Clear Day (magenta) and Shade pulse (gray) conditions, in units of $\mu$mol photons m$^{-2}$ s$^{-1}$. Experimental setup is displayed in the lower plot. Cells were grown for 12 hr under Clear Day conditions (dashed magenta lines), followed by 12 hr of darkness (dark gray boxes) for three days, and then exposed to a Shade pulse (dashed gray lines) at 8 hr after dawn during the fourth light period for one hour before being returned to Low Light conditions. Cells were sampled immediately before, during, and after the High Light pulse (indicated by arrows above plot). (C) Gene expression dynamics of dusk genes ($n$=281) under High Light pulse conditions. Gene expression is quantified as the log$_2$ fold change from the average expression of the gene over all time points in the Low Light condition (see Materials and methods, RNA sequencing for more details; data available in *Figure 3—source data 1*). Light intensity at each time point in the High Light pulse condition is indicated in a grayscale heat map next to the corresponding time point. (D) Gene expression dynamics of dusk genes ($n$=281) under Shade pulse conditions, plotted as in (C). Genes are ordered the same in (C) and (D), sorted by phase under Constant Light conditions (*Vijayan et al., 2009*). Data for the representative dusk gene *Synpcc7942_1567* is indicated by arrows in (C) and (D). (E) Gene expression dynamics of the representative dusk gene *Synpcc7942_1567* under Low Light (black) and High Light pulse (orange) conditions (left y-axis). The light profile for each condition is plotted as dashed lines of the same color with values corresponding to the right y-axis. (F) Gene expression dynamics of the representative dusk gene *Synpcc7942_1567* under Clear Day (magenta) and Shade pulse (gray) conditions, plotted as in (E). (G) Correlation between change in dusk gene expression and the change in enrichment of RNAP upstream of that gene after rapid changes in light intensity. The change in gene expression of a dusk gene (x-axis) and the corresponding change in RNAP enrichment upstream of that gene (y-axis) from the original condition after 60 min in High Light (orange triangles) or Shade (gray circles), plotted for the 82 dusk genes with detectable RNAP peaks in their promoters. See Materials and methods, ChIP-seq analysis for more details. Data is available in *Figure 3—source data 2*. The correlation coefficient between change in RNAP enrichment and change in downstream gene expression for the High Light and Shade conditions is indicated above the plot. (H) Regulation of RNAP recruitment to dusk genes by changes in light intensity. Increases in light intensity tend to repress the recruitment of RNAP to dusk genes to repress dusk gene expression (High Light pulse, Clear Day - midday), while decreases in light intensity (Shade pulse, Sunset of the Clear Day) tend to promote the recruitment of RNAP to dusk genes to activate their expression.

DOI: https://doi.org/10.7554/eLife.32032.010

The following source data and figure supplements are available for figure 3:

**Source data 1.** Normalized gene expression in High Light pulse and Shade pulse conditions.
DOI: https://doi.org/10.7554/eLife.32032.013
**Source data 2.** List of RNAP peaks, gene targets, and quantification of enrichment under High Light pulse and Shade pulse conditions.
DOI: https://doi.org/10.7554/eLife.32032.014
**Figure supplement 1.** Rapid changes in light intensity affect dawn gene expression in an opposite direction compared to dusk gene expression.
DOI: https://doi.org/10.7554/eLife.32032.011
**Figure supplement 2.** Changes in RNAP enrichment and downstream dusk gene expression after rapid changes in light intensity.
DOI: https://doi.org/10.7554/eLife.32032.012

dusk gene expression downstream of RpaA~P. Interestingly, ChIP-seq showed that light intensity fluctuations alter RpaA~P binding upstream of dusk genes (*Figure 4C*; *Figure 4—figure supplement 2*) in conjunction with RNAP binding upstream of the same gene (*Figure 4D*; *Figure 4—figure supplement 3*). The binding of RpaA~P and RNAP correlated with changes in downstream dusk gene expression (*Figure 4C*; *Figure 4—figure supplement 2*). Thus, light fluctuations control the binding of RpaA~P and RNAP to promoters, suggesting that light-induced changes in the binding of these factors may modulate the activation of dusk gene expression (*Figure 4E*).

Interestingly, RpaA regulation at a small number (~10) of promoters including that of *kaiBC* is not substantially affected by light intensity (*Figure 4C,D* - points around origin; *Figure 4—figure supplement 4*), demonstrating that the light-dependent regulation of RpaA binding is locus-specific. The KaiABC clock regulates RpaA~P levels independent of changes in light intensity (*Figure 4A,B*), and *kaiBC* gene expression dynamics do not substantially change in the Clear Day conditions compared to the Low Light condition (*Figure 4—figure supplement 4H–K*). Hence, the stabilizing PTO/transcription-translation feedback loop circadian circuit is robust to natural fluctuations in sunlight. The circadian clock can thus control gene expression independent of environmental changes at select promoters. It is possible that RpaA~P binding to some promoters is dependent on the association of RNAP with that promoter. As such, regulation that affects RNAP binding to a specific promoter, such as that by sigma factor activity (*Gruber and Gross, 2003*), could affect RpaA~P binding to select promoters.

Our analysis so far has established that the previous model for the regulation and expression of circadian genes in Constant Light conditions (*Figure 1A*) becomes more complex in natural

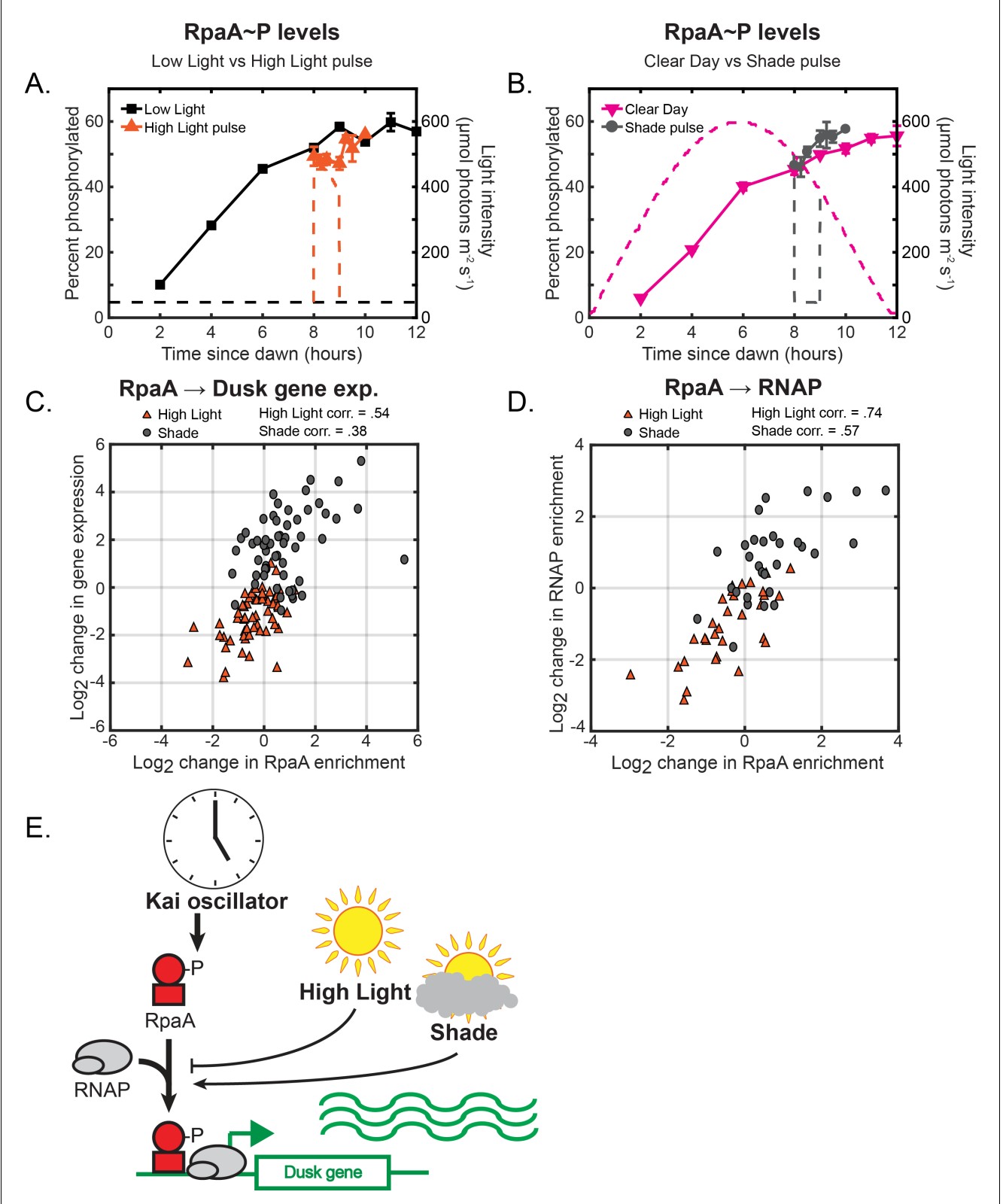

**Figure 4.** Changes in environmental light intensity regulate RpaA~P DNA binding activity and RNAP recruitment to control dusk gene expression downstream of clock regulation of RpaA. (A) Phosphorylation dynamics of RpaA under Low Light vs High Light pulse. Relative levels of phosphorylated RpaA were measured using Phos-tag Western blotting (left y-axis) in cells grown under Low Light conditions (black squares, see *Figure 2A* for Setup) or High Light pulse conditions (orange triangles, see *Figure 3A* for Setup). Each point represents the average of values measured in two independent

*Figure 4 continued on next page*

*Figure 4 continued*

Western blots, with error bars displaying the range of the measured values. See Materials and methods, Measurement of RpaA~P and RpaB~P levels for more details. Data is available in *Figure 4—source data 1*. The light profile for each condition is plotted as dashed lines of the same color with values corresponding to the right y-axis. (B) Phosphorylation dynamics of RpaA under Clear Day (magenta triangles, see *Figure 2A* for Setup) vs Shade pulse (gray circles, see *Figure 3B* for Setup) conditions, measured and plotted as in (A). (C) The change in enrichment of RpaA at a given peak upstream of a dusk gene (x-axis) and the corresponding change in expression of the downstream dusk gene (y-axis) from the original condition after 60 min in High Light (orange triangles) or Shade (gray circles), plotted for the 56 dusk genes with detectable RpaA peaks in their promoters. The correlation coefficient for the data taken in High Light and Shade conditions is indicated above the plot. See Materials and methods, ChIP-seq analysis for more details. Data is available in *Figure 4—source data 2*. (D) The change in enrichment of RpaA at a given peak upstream of a dusk gene (x-axis) and the corresponding change in RNAP enrichment upstream of the same gene (y-axis) from the original condition after 60 min in High Light (orange triangles) or Shade (gray circles), plotted for the 33 dusk genes with detectable RpaA and RNAP peaks in their promoters. The correlation coefficient for High Light and Shade data is indicated above the plot. See Materials and methods, ChIP-seq analysis for more details. (E) Model of regulation of dusk genes by RpaA under naturally-relevant conditions. The Kai PTO controls levels of RpaA~P independent of changes in environmental light intensity. Changes in light intensity regulate the recruitment of RpaA~P with RNAP to dusk genes to control their expression in response to environmental perturbations.

DOI: https://doi.org/10.7554/eLife.32032.015

The following source data and figure supplements are available for figure 4:

**Source data 1.** Quantification of relative RpaA~P levels.
DOI: https://doi.org/10.7554/eLife.32032.020
**Source data 2.** List of RpaA peaks, gene targets, and quantification of enrichment under High Light pulse and Shade pulse conditions.
DOI: https://doi.org/10.7554/eLife.32032.021
**Figure supplement 1.** Representative Western blots used to quantify relative levels of RpaA~P under dynamic light conditions.
DOI: https://doi.org/10.7554/eLife.32032.016
**Figure supplement 2.** Changes in RpaA enrichment and downstream dusk gene expression after rapid changes in light intensity.
DOI: https://doi.org/10.7554/eLife.32032.017
**Figure supplement 3.** Changes in RpaA and RNA polymerase enrichment upstream of dusk genes after rapid changes in light intensity.
DOI: https://doi.org/10.7554/eLife.32032.018
**Figure supplement 4.** Multifactorial behavior of RpaA~P at select promoters under changes in light intensity.
DOI: https://doi.org/10.7554/eLife.32032.019

environmental conditions, suggesting the involvement of other pathways. Thus, we next asked whether RpaB plays a role in controlling light-dependent expression of circadian genes. We observed that levels of RpaB~P changed rapidly in a direction opposite to the change in light (*Figure 5A,B*; *Figure 5—figure supplement 1*), suggesting that light affects RpaB activity through its phosphorylation state (*Figure 5E*). Levels of RpaB~P decreased ~3.1 fold after 15 min in the High Light pulse, and increased ~1.9 fold after 15 min in the Shade pulse, concomitant with the rapid repression and induction of many dusk genes (*Figure 3*). Further, RpaB~P levels increased ~1.7 fold between 10 and 12 hr after dawn in the Clear Day condition concomitant with the decrease in light during Sunset and the strong induction of many dusk genes (*Figure 2*). This strong correlation between RpaB~P levels and the expression of dusk genes under dynamic light conditions (also compare *Figure 3E,F* to *Figure 5A,B*) suggests that RpaB~P acts as an activator of dusk gene expression. Indeed, using ChIP-seq we found that RpaB binds upstream of a large subset of dusk genes (42/281 dusk genes, *Figure 5—source data 2*). RpaB binding upstream of these genes shifts after rapid changes in light (*Figure 5C*; *Figure 5—figure supplement 2*), correlating with changes in RpaB~P levels (*Figure 5A,B*), RNAP binding upstream of the same gene (*Figure 5D*; *Figure 5—figure supplement 3*), and downstream dusk gene expression (*Figure 5C*; *Figure 5—figure supplement 2*). These results suggest that RpaB~P directly activates the expression of many dusk genes by binding to promoters with RNAP (*Figure 5E*). Thus, changes in sunlight can regulate dusk genes by adjusting RpaB~P levels (*Figure 5E*).

Because RpaA and RpaB bind only a subset of light-responsive dusk genes (*Figure 6A,B*), additional factors must be involved in controlling light-responsive dusk gene expression. Sigma factors are sequence-specific RNAP subunits which regulate gene expression in bacteria (*Gruber and Gross, 2003*). Interestingly, RpaA, RpaB, and RNAP bind to the promoters of three sigma factor genes (*Figure 6C*; *Figure 6—figure supplement 1A–C*). The binding of RpaA, RpaB, and RNAP to these promoters shifts in conjunction after abrupt changes in light intensity, correlating with light-responsive changes in expression of these genes (*Figure 6D*; *Figure 6—figure supplement 1D–F*).

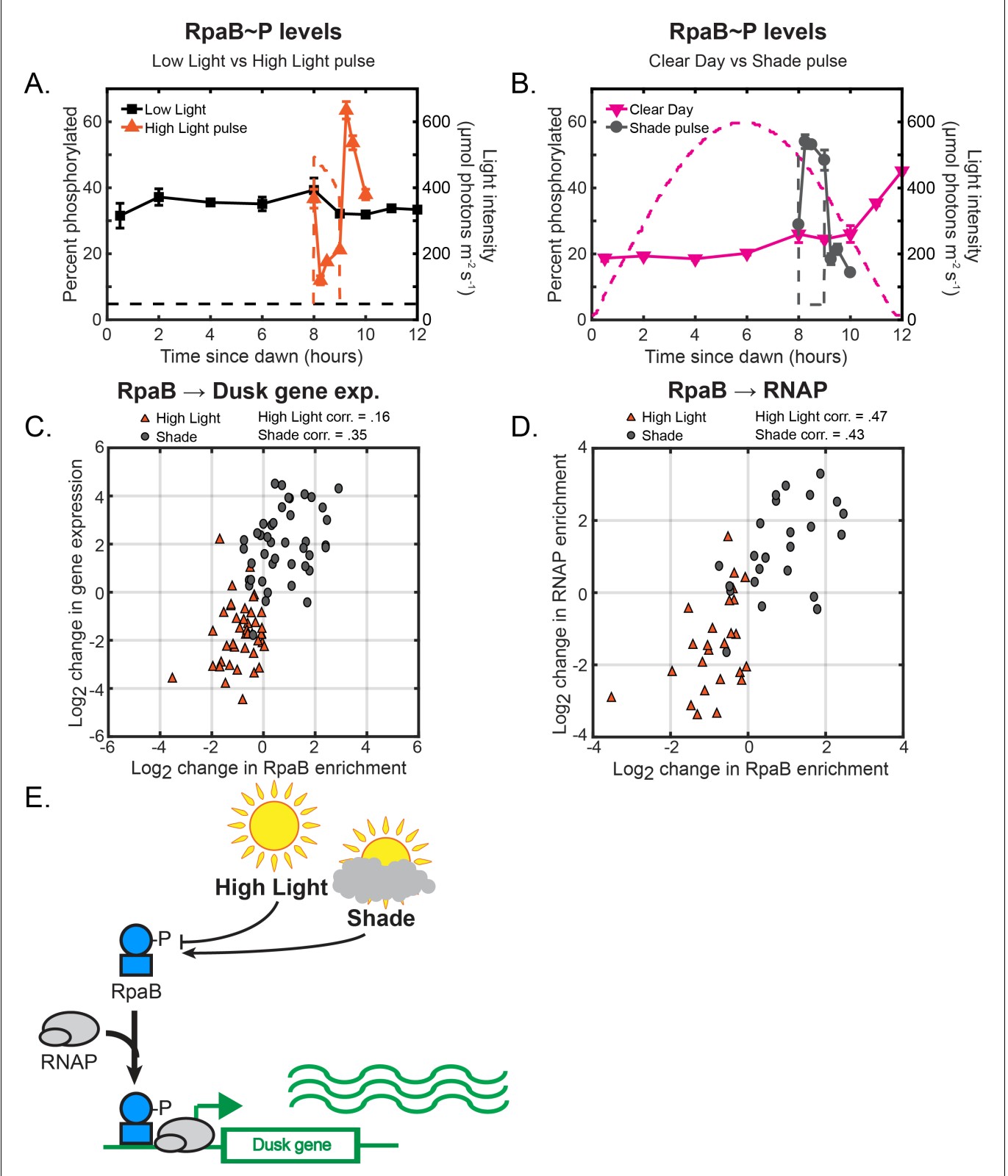

**Figure 5.** Light-induced changes in RpaB ~ P levels modulate RpaB and RNAP binding upstream of dusk genes to directly regulate dusk gene expression in response to light. (**A**) Phosphorylation dynamics of RpaB under Low Light vs High Light pulse. Relative levels of phosphorylated RpaB were measured using Phos-tag Western blotting (left y-axis) in cells grown under Low Light conditions (black squares, see *Figure 2A* for Setup) or High Light pulse conditions (orange triangles, see *Figure 3A* for Setup). Each point represents the average of values measured in two independent Western

*Figure 5 continued on next page*

*Figure 5 continued*

blots, with error bars displaying the range of the measured values. See Materials and methods, Measurement of RpaA ~ P and RpaB ~ P levels for more details. Data is available in *Figure 5—source data 1*. The light profile for each condition is plotted as dashed lines of the same color with values corresponding to the right y-axis. (B) Phosphorylation dynamics of RpaB under Clear Day (magenta triangles, see *Figure 2A* for Setup) vs Shade pulse (gray circles, see *Figure 3B* for Setup) conditions, measured and plotted as in (A). (C) The change in enrichment of RpaB at a given peak upstream of a dusk gene (x-axis) and the corresponding change expression of the downstream dusk gene (y-axis) from the original condition after 60 min in High Light (orange triangles) or Shade (gray circles), plotted for the 42 dusk genes with detectable RpaB peaks in their promoters. The correlation coefficient for High Light and Shade data is indicated above the plot. See Materials and methods, ChIP-seq analysis for more details. Data is available in *Figure 5—source data 2*. (D) The change in enrichment of an RpaB at a given peak upstream of a dusk gene (x-axis) and the corresponding change in RNAP enrichment upstream of the same gene (y-axis) from the original condition after 60 min in High Light (orange triangles) or Shade (gray circles), plotted for the 27 dusk genes with detectable RpaB and RNAP peaks in their promoters. he correlation coefficient for High Light and Shade data is indicated above the plot. See Materials and methods, ChIP-seq analysis for more details. (E) Model of regulation of dusk genes by RpaB under naturally-relevant conditions. Changes in light regulate RpaB ~ P levels. RpaB ~ P binds with RNAP to dusk genes to control their expression in response to environmental perturbations.

DOI: https://doi.org/10.7554/eLife.32032.022

The following source data and figure supplements are available for figure 5:

**Source data 1.** Quantification of relative RpaB~P levels.

DOI: https://doi.org/10.7554/eLife.32032.026

**Source data 2.** List of RpaB peaks, gene targets, and quantification of enrichment under High Light pulse and Shade pulse conditions.

DOI: https://doi.org/10.7554/eLife.32032.027

**Figure supplement 1.** Representative Western blots used to quantify relative levels of RpaB ~ P under dynamic light conditions.

DOI: https://doi.org/10.7554/eLife.32032.023

**Figure supplement 2.** Changes in RpaB enrichment and downstream dusk gene expression after rapid changes in light intensity.

DOI: https://doi.org/10.7554/eLife.32032.024

**Figure supplement 3.** Changes in RpaB and RNA polymerase enrichment upstream of dusk genes after rapid changes in light intensity.

DOI: https://doi.org/10.7554/eLife.32032.025

These sigma factor genes show light-dependent dusk gene expression patterns (*Figure 6—figure supplement 1G–L*) that mirror those of the larger group of dusk genes (*Figures 2* and *3*), suggesting that these sigma factors could regulate the expression of other dusk genes. Thus, RpaA and RpaB may indirectly regulate the expression of non-target dusk genes by controlling the circadian and light-responsive expression of sigma factor genes (*Hanaoka et al., 2012*), similar to how RpaA drives all dusk gene expression in Constant Light conditions by binding to a subset of dusk genes (*Markson et al., 2013*). It is also possible that changes in light intensity affect dusk gene expression in a manner independent of RpaA ~ P and RpaB ~ P regulation. For instance, global growth-rate-dependent gene regulatory mechanisms such as the stringent response (*Scott et al., 2010*; *Burnap, 2015*; *Ryals et al., 1982*; *Hood et al., 2016*) likely cause some of the light-dependent changes in circadian gene expression due to unavoidable differences in the growth rate in different light conditions (*Figure 2—figure supplement 1*).

We have defined a regulatory picture in which changes in light intensity affect the activity of RpaA and RpaB to control the expression of dusk genes. However, light affects RpaA activity in complex and promoter-specific ways. Additionally, light-dependent regulation in addition to that mediated by RpaA and RpaB may control dusk gene expression in response to environmental perturbations. Still, despite the apparent complexity of regulation of dusk genes in response to light fluctuations, the expression of almost all dusk genes show strikingly regular dynamics (*Figures 2* and *3*). Furthermore, the activity of RpaA and RpaB at a subset of promoters (especially those of sigma factor genes) could lead to pervasive and coordinated changes in the expression of other dusk genes. Hence, we reasoned that mathematical models (*Alon, 2006*) of RpaA and RpaB activity might effectively describe the regulatory circuits underlying the dynamics of large groups of dusk genes. Such an approach would enable an understanding of the basic principles of interaction between circadian gene expression regulation with light-dependent regulation without needing to describe all underlying molecular mechanisms.

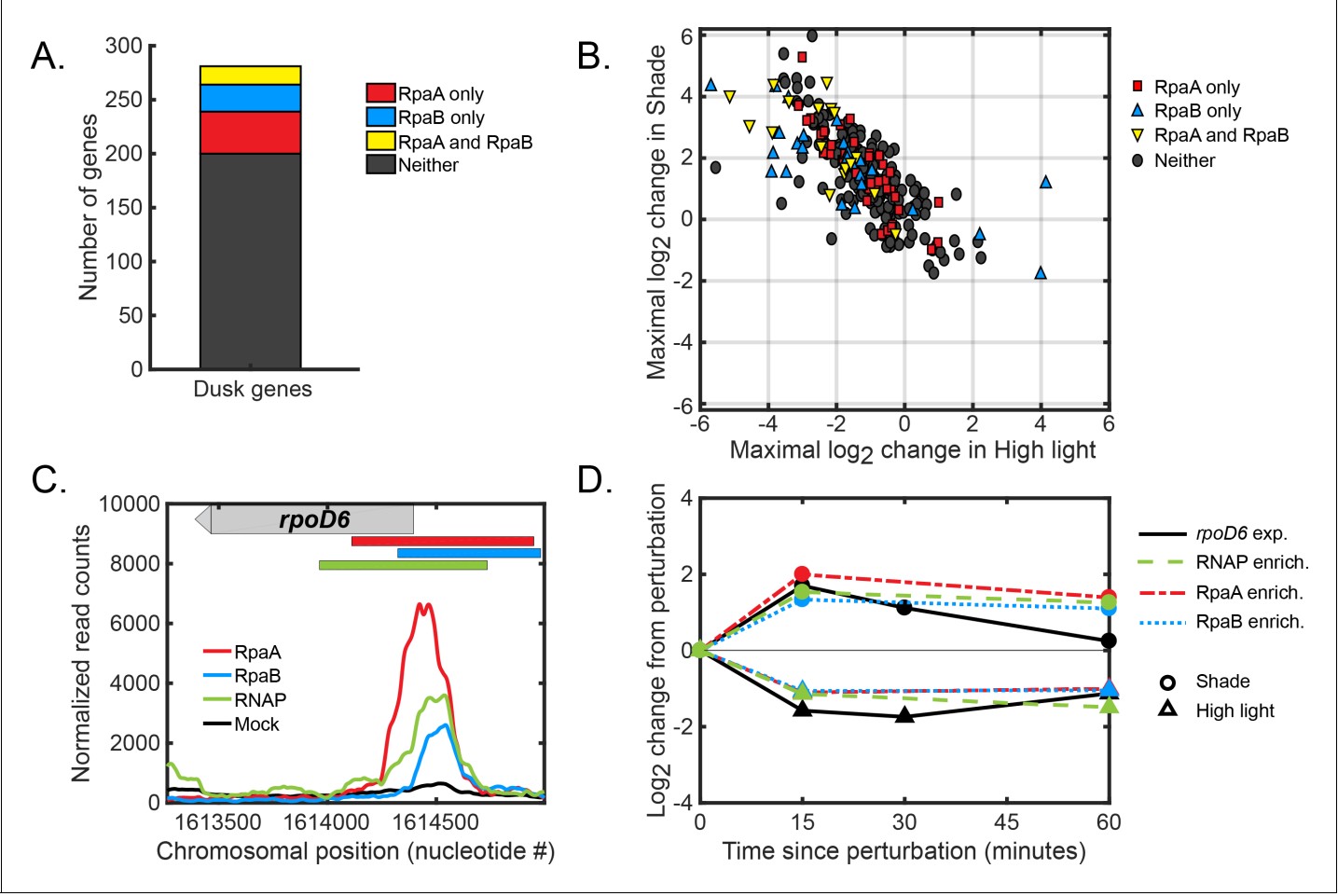

**Figure 6.** Global regulation of dusk gene expression in response to light changes. (**A**) Number of dusk gene targets of RpaA only (red), RpaB only (blue), RpaA and RpaB (yellow), or neither (black). Target genes of binding sites of RpaA and RpaB were determined using chromatin immunoprecipitation followed by sequencing under several different light conditions (see Materials and methods, ChIP-seq analysis, for more details. See *Figure 4—source data 2* or *Figure 5—source data 2* for full lists of RpaA and RpaB peaks associated with dusk genes). (**B**) Light-responsive changes in gene expression of dusk genes. For each dusk gene, we calculated the maximal $log_2$ change in expression during the High Light pulse (x-axis) or Shade pulse (y-axis) from 8 hr since dawn in the Low light or Clear day conditions, respectively, using the data from *Figure 3*. (**C**) Normalized ChIP-seq signal of RpaA (red), RpaB (blue), RNAP (green) and mock IP (black) upstream of the dusk sigma factor gene *rpoD6* at 8 hr since dawn in Low Light. The chromosomal location of the gene is located on the plot with a gray bar with an arrow indicating directionality of the gene. The location of RpaA, RpaB, and RNAP peaks are indicated on top of the plot with red (RpaA), blue (RpaB), and green (RNAP) bars. See Materials and methods, ChIP-seq analysis for more details. (**D**) Changes in enrichment upstream of *rpoD6* of RpaA (red), RpaB (blue), and RNAP (green) and changes in *rpoD6* gene expression (black) after exposure to the High Light pulse (triangles) or the Shade pulse (circles). See Materials and methods, ChIP-seq analysis for more details.

DOI: https://doi.org/10.7554/eLife.32032.028

The following figure supplement is available for figure 6:

**Figure supplement 1.** Regulation of dusk sigma factor gene expression by RpaA and RpaB.
DOI: https://doi.org/10.7554/eLife.32032.029

## Phenomenological models suggest simple principles underlying the activation of clusters of light-responsive dusk genes

We find that dusk genes collectively display a small number of responses to changes in environmental light intensity. Using k-means clustering of the gene expression dynamics from our different light profiles (*Figures 2* and *3*), as well as from perturbations of RpaA (*Figure 2—figure supplement 2* [*Markson et al., 2013*]), we identify three major groups of dusk genes (35–80 genes, see *Figure 7—source data 1* for full lists) which show distinct and coordinated changes in gene expression over

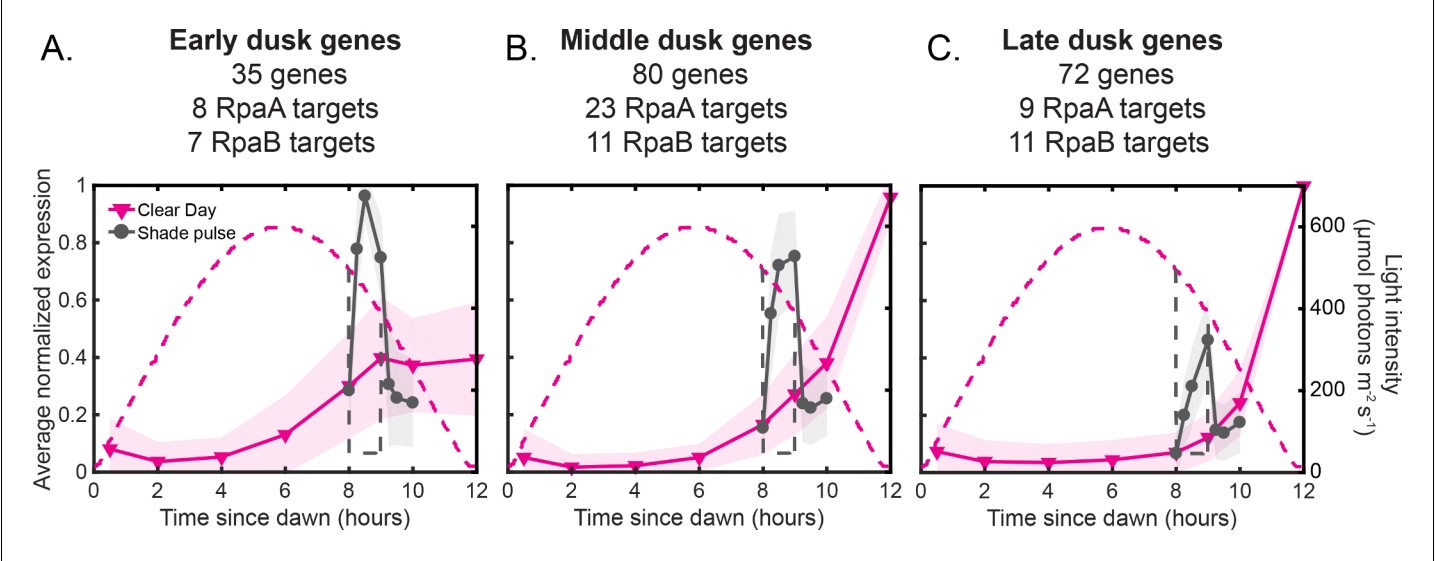

**Figure 7.** Dusk genes group into three major clusters that show distinct and coordinated responses to changes in light intensity. (A) Average expression profiles of genes belonging to the Early dusk gene cluster under Clear Day (magenta) and Shade pulse (gray) conditions (left y-axis). Dusk genes were grouped using k-means clustering of their normalized expression dynamics in response to the four light conditions of this study and perturbations of RpaA activity in Constant Light conditions (*Figure 7—figure supplement 1*, [*Markson et al., 2013*]), and clusters were named based on their order of activation. See Materials and methods - K-means clustering for more details, and *Figure 7—source data 1* for full lists of genes in each cluster. The number of genes within the cluster, as well as the number of genes with an RpaA or RpaB peak in their promoters (targets) is listed. The expression values of each gene across all four light conditions in this work were normalized to a range of 0 to 1, and the normalized expression values were averaged within each cluster (solid lines). The shaded region on the plot indicates the standard deviation of the normalized expression values within the cluster. The light intensity profile for each condition is plotted as dashed lines in the same color with values corresponding to the right y-axis. (B) Average expression profiles of genes belonging to the Middle dusk gene cluster under Clear Day (magenta) and Shade pulse (gray) conditions (left y-axis), presented as in (A). (C) Average expression profiles of genes belonging to the Late dusk gene cluster under Clear Day (magenta) and Shade pulse (gray) conditions (left y-axis), presented as in (A).

DOI: https://doi.org/10.7554/eLife.32032.030

The following source data and figure supplement are available for figure 7:

**Source data 1.** Lists of genes belonging to the Early, Middle, and Late dusk clusters, and scaled gene expression values.
DOI: https://doi.org/10.7554/eLife.32032.032

**Figure supplement 1.** Average expression profiles of the major dusk gene clusters under various conditions.
DOI: https://doi.org/10.7554/eLife.32032.031

circadian time and in response to changes in light intensity (*Figure 7*; *Figure 7—figure supplement 1*). Under Constant Light conditions, all three clusters are activated by RpaA ~ P but display distinct activation dynamics from dawn to dusk (*Figure 7—figure supplement 1B*) that are mirrored under our Low Light conditions (*Figure 7—figure supplement 1A*). We named the clusters the Early, Middle, and Late dusk genes based on the order of activation.

The Shade pulse and Sunset in the Clear Day condition have differing effects on the expression of each of the major dusk gene clusters. Early dusk gene expression rapidly increases in response to Shade, but during Sunset plateaus at ~ 1/2 of the maximal gene expression reached in Shade (*Figure 7A*). Conversely, the Late gene cluster responds most strongly to Sunset in Clear Day conditions but has a mild increase in expression in Shade relative to the Early and Middle dusk genes (*Figure 7C*). In contrast, the Middle gene cluster is induced to a similar magnitude by both Shade and Sunset (*Figure 7B*). Shade and Sunset represent similar light changes that occur at different times of day (afternoon and dusk, respectively). As such, the Early and Late dusk gene clusters are differentially induced by a decrease in light intensity depending on the time of day in which it occurs. This circadian effect on the intensity and dynamics of a response to environmental change is a signature of circadian gating (*Hotta et al., 2007*; *Greenham and McClung, 2015*). Though circadian gating has been observed (e.g., [*Belbin et al., 2017*]) and modeled without any knowledge of the

transcriptional regulation (*Dalchau et al., 2010*) in plants, it remains unclear what gene regulatory circuits are sufficient to explain such behavior.

At present there is no mechanistic model to explain the differential response of these clusters to circadian regulation and changes in sunlight. Given that there are unknown regulators involved in circadian gene expression (*Figure 6A,B*), and because it is not possible to exhaustively test all possible models of regulation of dusk gene expression, we sought to construct the simplest models that can describe the expression dynamics of these clusters using a phenomenological modeling approach. Such models can be used to highlight regulatory architectures that are sufficient to recapitulate the observed gene expression dynamics, as well as direct further mechanistic studies to reveal the underlying molecular details of regulation.

Given the clear roles for RpaA~P and RpaB~P in activating dusk genes, we asked whether the dynamic expression of the major dusk gene clusters in naturally-relevant light conditions could be described by these variables. We constructed phenomenological models that describe the kinetics of the synthesis and breakdown of an average gene in each of the dusk gene clusters (*Mangan and Alon, 2003*) (see Materials and methods, Mathematical modeling). The rate of synthesis was the sum of a baseline rate of transcription and a maximal adjustable rate of transcription that could be modulated by the activity of one or more regulators. We described the the effects of a regulator such as RpaA~P or RpaB~P using a Hill function, whose shape is determined by the Hill coefficient and the coefficient of activation. We determined how well a model could describe the dynamics of a cluster by fitting it to the Clear Day and Shade pulse data and assuming all parameters could vary freely (see Materials and methods, Mathematical modeling; *Table 1*).

We began by asking whether levels of RpaA~P or RpaB~P (*Figure 8A*) can describe the gene expression dynamics of the major dusk clusters in natural light conditions. We first constructed models in which dusk cluster gene expression is solely dependent on RpaA~P. Activation by RpaA~P can recapitulate the ordered activation of the dusk gene clusters through differential coefficients of activation for RpaA~P, but cannot describe the light-responsive expression of these genes (RpaA-only models, *Figure 8—figure supplement 1A–C*; *Table 2*). Further, activation by RpaB~P alone cannot describe the dusk gene expression patterns of the clusters (RpaB-only models, *Figure 8—figure supplement 1D–F*; *Table 2*). However, models in which dusk gene expression is a function of BOTH RpaA~P and RpaB~P can recapitulate much of the time-of-day and light intensity dependent expression of the Early and Late clusters and nearly all of the expression dynamics of the Middle clusters (RpaA and RpaB models, *Figure 8B–E*; *Table 2*). This suggests that RpaB~P is a variable which can capture the effects of dynamic light conditions on RpaA~P activity. The fit parameters for simple joint activation can accommodate indirect activation through downstream regulators like sigma factors and thus do not require direct RpaA/B binding to all genes. Conceptually, our results suggest that transcription factors whose activity track the measured dynamics of both RpaA~P and RpaB~P can describe the circadian and light-responsive expression of dusk genes. However, joint activation by RpaA~P and RpaB~P predicts that the Early and Late clusters will respond similarly to Shade and Sunset in Clear Day conditions (*Figure 8C,E*), and thus cannot capture well the circadian gating of these clusters.

**Table 1.** Fitting bounds.

Bounds used for fitting the variables in our simple model of gene expression. H is the Hill coefficient, $\beta$ is the max transcription rate, $\alpha$ is the decay/dilution rate, $B$ is the background transcription rate, and $K$ is a coefficient of activation/repression (see *equations 1-3*, p. 1–3). The units of $\beta$, $\alpha$, and $B$ are normalized expression/hr; $K$ is in normalized expression units.

| Variable | Lower bound | Upper bound |
| --- | --- | --- |
| **H** | 0 | 7 |
| $\beta$ | 0 | 80 |
| $\alpha$ | 0 | 80 |
| $B$ | 0 | 10 |
| $K$ | 0 | 1 |

DOI: https://doi.org/10.7554/eLife.32032.038

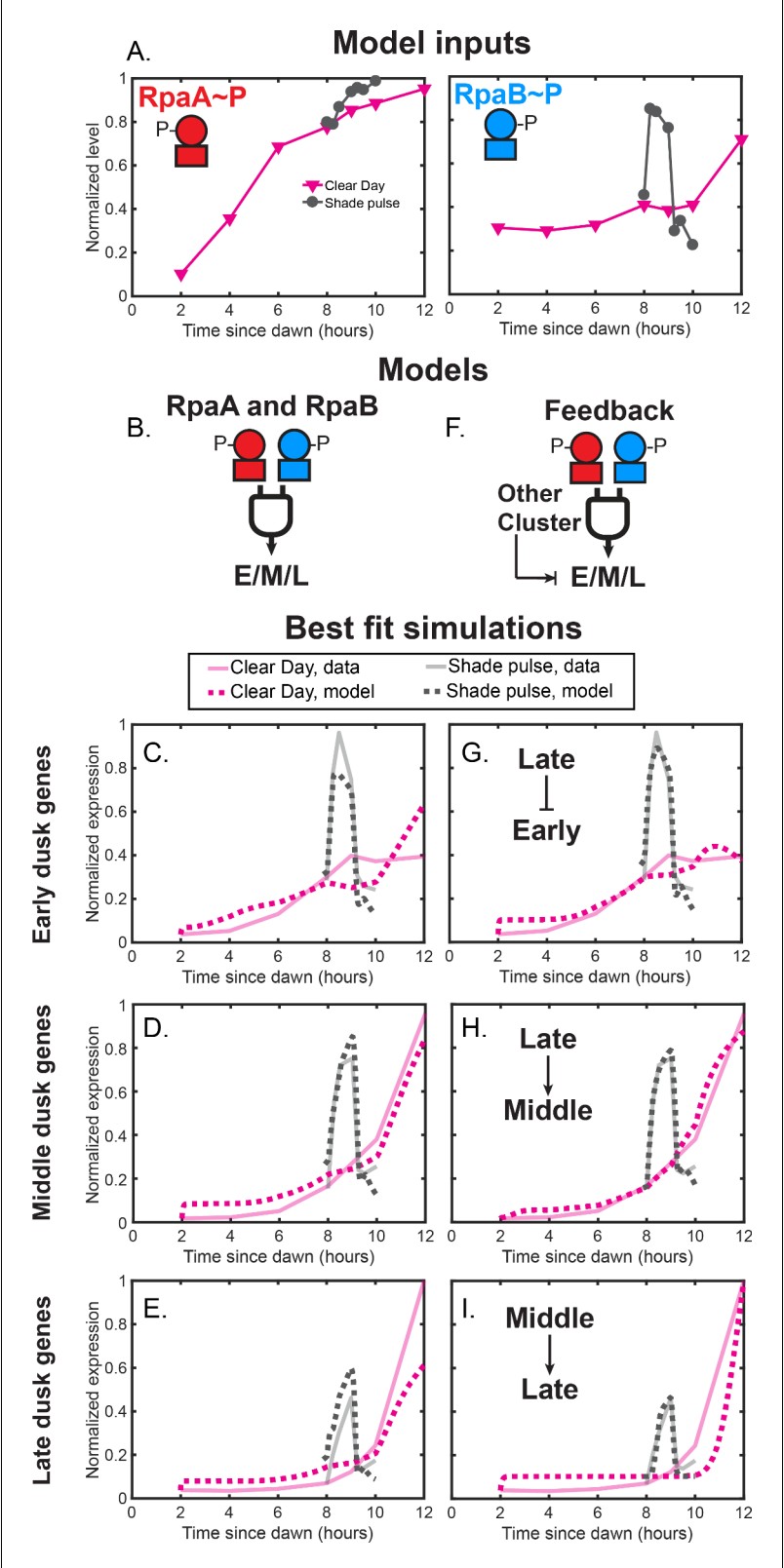

**Figure 8.** Phenomenological modeling of the activation of clusters of light-responsive dusk genes. (**A**) Normalized RpaA∼P levels (left plot) and RpaB∼P levels (right plot) under Clear Day (magenta) and Shade pulse (gray) conditions used as input for mathematical models of dusk gene expression. RpaA∼P or RpaB∼P levels from all four light conditions were normalized to a range of 0 to 1. (**B**) In the 'RpaA and RpaB' models, RpaA∼P and RpaB∼P jointly activate the expression of the Early (E), Middle (M), or Late (L) cluster. See Materials and methods - Mathematical modeling for

*Figure 8 continued on next page*

*Figure 8 continued*

more details. (C) Simulations of the best fit 'RpaA and RpaB' model for the Early dusk genes. Average cluster expression data is shown as faded solid lines, and the best fit simulations are shown as dotted lines. Data for Clear Day conditions are plotted in magenta, and Shade pulse in gray. See Materials and methods - Mathematical modeling for more details. (D) Simulations of the best fit 'RpaA and RpaB' model for the Middle dusk genes, plotted as in (C). (E) Simulations of the best fit 'RpaA and RpaB' model for the Late dusk genes, plotted as in (C). (F) In the 'Feedback' models, another cluster activates or represses the expression of the Early (E), Middle (M), or Late (L) cluster alongside joint activation by RpaA~P and RpaB~P. (G) Simulations of the best fit 'Feedback' model for the Early dusk genes, plotted as in (C). In this model, Late cluster expression represses Early cluster expression alongside activation by RpaA~P and RpaB~P. (H) Simulations of the best fit 'Feedback' model for the Middle dusk genes, plotted as in (C). In this model, Late cluster expression activates Middle cluster expression alongside activation by RpaA~P and RpaB~P. (I) Simulations of the best fit 'Feedback' model for the Late dusk genes, plotted as in (C). In this model, Middle cluster expression activates Late cluster expression alongside activation by RpaA~P and RpaB~P.

DOI: https://doi.org/10.7554/eLife.32032.033

The following figure supplements are available for figure 8:

**Figure supplement 1.** Best fit simulations of 'RpaA-only' and 'RpaB-only' models in which RpaA~P or RpaB~P solely activates the expression of the dusk gene clusters.

DOI: https://doi.org/10.7554/eLife.32032.034

**Figure supplement 2.** Models in which either the Middle or Late cluster feeds back to influence Early cluster expression.

DOI: https://doi.org/10.7554/eLife.32032.035

**Figure supplement 3.** Models in which either the Early or Late cluster feeds back to influence Middle cluster expression.

DOI: https://doi.org/10.7554/eLife.32032.036

**Figure supplement 4.** Models in which either the Early or Middle cluster feeds back to influence Late cluster expression.

DOI: https://doi.org/10.7554/eLife.32032.037

**Table 2.** Fitting results.

The definitions of the variables are given in **Equations 1-3**, p. 1–3. The error is defined as the square root of the sum of the squared deviations between simulation and data.

| Model | Cluster | Figure | $B_X$ | $\beta_X$ | $\alpha_X$ | $K_{AX}$ | $H_{AX}$ | $K_{BX}$ | $H_{BX}$ | $K_{YX}$ | $H_{YX}$ | Error |
|---|---|---|---|---|---|---|---|---|---|---|---|---|
| RpaA-only | Early | 7D | 0.71 | 37.54 | 72.71 | 0.71 | 6.76 | - | - | - | - | 0.85 |
| RpaB-only | Early | 7-Fig. Supp. 2C | 0.37 | 24.03 | 78.62 | - | - | 0.37 | 0.78 | - | - | 1.01 |
| RpaA and RpaB | Early | 7G | 0.35 | 51.28 | 37.76 | 0.35 | 4.19 | 0.8 | 2.5 | - | - | 0.41 |
| Feedback, M act. | Early | 7-Fig. Supp. 3A | 0.01 | 55.85 | 30.01 | 0.01 | 0.3 | 0.87 | 2.38 | 0.06 | 2.47 | 0.37 |
| Feedback, M rep. | Early | 7-Fig. Supp. 3B | 0.67 | 58.69 | 38.89 | 0.67 | 6.96 | 0.62 | 2.47 | 0.96 | 7 | 0.24 |
| Feedback, L act. | Early | 7-Fig. Supp. 3C | 0.2 | 35.87 | 19.03 | 0.2 | 4.43 | 0.98 | 3.35 | 0.05 | 6.15 | 0.38 |
| Feedback, L rep. | Early | 7I, 7-Fig. Supp. 3D | 0.75 | 69.34 | 42.68 | 0.75 | 6.22 | 0.59 | 3.53 | 0.71 | 2.39 | 0.21 |
| RpaA-only | Middle | 7D | 0.79 | 37.95 | 63 | 0.79 | 6.76 | - | - | - | - | 0.86 |
| RpaB-only | Middle | 7-Fig. Supp. 2C | 0.26 | 0.03 | - | - | - | 0.26 | 5.6 | - | - | 0.85 |
| RpaA and RpaB | Middle | 7G | 1 | 57.46 | 25.97 | 1 | 4.96 | 0.52 | 4.12 | - | - | 0.29 |
| Feedback, E act. | Middle | 7-Fig. Supp. 4A | 0.8 | 23.73 | 22.19 | 0.8 | 6.96 | 0.49 | 4.53 | 0.21 | 6.35 | 0.32 |
| Feedback, E rep. | Middle | 7-Fig. Supp. 4B | 0.73 | 71.08 | 39.24 | 0.73 | 5.14 | 0.53 | 6.58 | 0.74 | 0.88 | 0.35 |
| Feedback, L act. | Middle | 7I, 7-Fig. Supp. 4C | 0.18 | 78.63 | 76.5 | 0.18 | 6.09 | 0.33 | 2.64 | 0.16 | 1.55 | 0.16 |
| Feedback, L rep. | Middle | 7-Fig. Supp. 4D | 0.68 | 31.02 | 17.98 | 0.68 | 3.34 | 0.57 | 6.79 | 1 | 0 | 0.44 |
| RpaA-only | Late | 7D | 0.96 | 39.82 | 64.37 | 0.96 | 6.7 | - | - | - | - | 0.78 |
| RpaB-only | Late | 7-Fig. Supp. 2C | 0.05 | 0 | 0 | - | - | 0.05 | 0.68 | - | - | 0.79 |
| RpaA and RpaB | Late | 7G | 0.95 | 77.65 | 67.1 | 0.95 | 7 | 0.48 | 5.9 | - | - | 0.5 |
| Feedback, E act. | Late | 7-Fig. Supp. 5A | 0.99 | 23.93 | 20.01 | 0.99 | 5.8 | 0.4 | 6.95 | 0.18 | 6.77 | 0.53 |
| Feedback, E rep. | Late | 7-Fig. Supp. 5B | 0.76 | 59.81 | 18.43 | 0.76 | 6.22 | 0.69 | 6.13 | 0.47 | 3.12 | 0.29 |
| Feedback, M act. | Late | 7I, 7-Fig. Supp. 5C | 0.37 | 27.3 | 16.09 | 0.37 | 3.72 | 0.01 | 3.46 | 0.91 | 6.23 | 0.22 |
| Feedback, M rep. | Late | 7-Fig. Supp. 5D | 0.86 | 25.1 | 14.46 | 0.86 | 6.92 | 0.48 | 7 | 1 | 0 | 0.52 |

DOI: https://doi.org/10.7554/eLife.32032.039

We reasoned that additional regulatory interactions downstream of RpaA and RpaB, or 'network motifs' (*Alon, 2006*), could account for the observed gating of the Early and Late clusters. Thus, we constructed models in which dusk cluster gene expression is positively or negatively dependent on the expression of another cluster alongside activation by RpaA~P and RpaB~P (Feedback models, *Figure 8F–I*; *Figure 8—figure supplements 2–4*; *Table 2*). Interestingly, the gating of the Early cluster is recapitulated by a model incorporating an incoherent feedforward loop in which the Late cluster represses Early cluster expression downstream of RpaA~P and RpaB~P activation (*Figure 8G*; *Figure 8—figure supplement 2*; *Table 2*). Further, the gating of the Late cluster is well described by a coherent feedforward loop in which Late cluster expression is dependent on RpaA~P, RpaB~P, AND Middle cluster expression levels (*Figure 8I*; *Figure 8—figure supplement 4*; *Table 2*). Thus, we highlight regulatory schemes downstream of RpaA and RpaB which can generate large time-of-day differences, or circadian gating, in the response to a decrease in light intensity.

Our results highlight that the measured dynamics of RpaA~P and RpaB~P can account for the dynamics of large groups of clock-controlled genes after environmental changes and suggest regulatory schemes that can diversify gene expression responses downstream of RpaA and RpaB. The models suggested here offer constraints and testable hypotheses to guide future studies of the molecular mechanisms underlying these responses.

## Discussion

### Changes in light adjust circadian gene expression to optimize metabolism in response to shifting ambient light intensity

We show that natural fluctuations in light intensity significantly affect the dynamics of circadian gene expression (*Figures 2* and *3*). While previous studies have measured genome-wide gene expression in a single natural light condition (*Waldbauer et al., 2012*), here we compare genome-wide circadian gene expression in several physiologically-relevant conditions, including Clear Day, High Light pulse, Shade pulse, and Low Light, to carefully dissect the effects of light on clock output. Natural light changes most greatly affected a large fraction of the dusk genes (*Figures 2B* and *3C, D*), possibly because most of the direct targets of RpaA are dusk genes (*Markson et al., 2013*). We speculate that the opposing trends we observe in dawn gene expression (*Figure 2—figure supplement 3* and *Figure 3—figure supplement 1*) may in part be due to competition for RNAP between the dusk and dawn genes (*Gruber and Gross, 2003*; *Mauri and Klumpp, 2014*) or by growth-rate-dependent mechanisms (*Scott et al., 2010*), as this group of genes contains the primary growth genes. A systematic exploration of the effects of light on circadian genes will be necessary to fully elaborate the contributions of light, clock, and growth rate on circadian gene dynamics.

We find that large groups of light-responsive dusk genes are activated by diminished light conditions to different extents depending on the time of day the stimulus is applied. These differences in activation may serve to optimally change metabolism for a given light condition and time of day. The light-responsive dusk genes grouped into three clusters - Early, Middle, and Late - with different activation dynamics during Sunset at the end of the Clear Day versus the Shade pulse in the afternoon (*Figure 7*, see *Figure 7—source data 1* for full lists of genes in each cluster). Glycogen breakdown genes and the central carbon metabolism genes glyceraldehyde-3-phosphate dehydrogenase and oxalate decarboxylase belong to the Middle dusk genes, which are activated to similar levels by Shade and Sunset (*Figure 7B*). This suggests that cyanobacteria delay the activation of glycogen breakdown pathways (*Reimers et al., 2017*) until just before dusk when grown under Clear Day conditions, but can transiently activate these genes in response to Shade to access alternate energy reserves if necessary. Interestingly, genes encoding pyridine nucleotide transhydrogenase, which reversibly converts NADH to the NADPH required for electron transport, belong to the Late cluster and are strongly activated only by Sunset and not afternoon Shade (*Figure 7C*). Such a response might delay the adjustment of the relative levels of NADH/NADPH until only when absolutely needed at night, when NADPH is potentially important for defense against reactive oxygen species (*Diamond et al., 2017*). The cytochrome c oxidase genes belong to the Early cluster, which respond more intensely to Shade than to Sunset (*Figure 7A*). This enzyme is essential for preventing photo-damage in response to rapid changes in light intensity (*Lea-Smith et al., 2013*); such changes are not expected to occur during the night, where it serves solely as the terminal electron acceptor for

respiration. More generally, the genome-wide gene expression dynamics measured here qualitatively agree with predictions from a whole-cell model of *S. elongatus* that assumed optimization of growth (*Reimers et al., 2017*). To resolve how the circadian and light-dependent transcriptional changes effect these metabolic changes, future studies must measure enzyme levels and metabolic fluxes under fluctuating light conditions.

## Mechanistic principles underlying the activation of light-responsive dusk genes

While light does not alter the post-translational oscillator/transcription-translation feedback loop circadian circuit, it regulates the activation of dusk genes via RpaA~P promoter binding (*Figure 4*) and RpaB promoter binding through its phosphorylation state (*Figure 5*) at a subset of dusk genes. RpaA binding upstream of its target genes under dynamic light conditions (*Figure 4C*) correlates with the changes in expression of non-RpaA target genes (*Figure 6B*). Thus, RpaA~P may remain the 'master regulator' of circadian gene expression whose promoter binding activity is altered by other molecular factors that encode information about the environment, such as RpaB. Previous work suggested that changes in RpaB~P phosphorylation would alter RpaA~P levels through competition with the enzymes that control RpaA~P levels (*Espinosa et al., 2015*). However, we find that RpaA~P levels remain constant (*Figure 4A,B*) under conditions in which RpaB~P levels change substantially (*Figure 5A,B*), arguing that RpaB~P does not influence RpaA~P levels. RpaB~P might influence RpaA~P binding at promoters where both proteins bind (*Figure 6—figure supplement 1*) as previously suggested (*Hanaoka et al., 2012*), and joint control of sigma factors by RpaA and RpaB could feedback to affect RpaA binding at select promoters. Still, the question of how light changes RpaA~P binding in a promoter-specific way remains unclear.

We define a clear role for the stress-responsive transcription factor RpaB as a transcriptional activator of a large subset of dusk genes (*Figure 5E*). Further, we demonstrate that decreases in light intensity like a Shade Pulse lead to increases in RpaB~P levels to allow RpaB to activate the expression of genes. This result shows that RpaB acts in scenarios beyond its previously appreciated role in High Light stress (*Kato et al., 2011*; *Seki et al., 2007*; *Hanaoka and Tanaka, 2008*; *López-Redondo et al., 2010*). RpaA~P and RpaB~P might cooperate to indirectly regulate the expression of most light-responsive dusk genes by jointly controlling the expression levels of multiple sigma factors (*Figure 6—figure supplement 1*) (*Hanaoka et al., 2012*). However, our attempts to cleanly perturb RpaB activity to further explore its role as a regulator of dusk genes were unsuccessful, in part because the *rpaB* gene is essential (*López-Redondo et al., 2010*). The role of sigma factors in this network of regulation, while strongly implied, remains ambiguous and attempts to assess this role using genetic deletion of sigma factors yielded inconclusive results. More subtle approaches such as anchors away (*Haruki et al., 2008*) might allow perturbation experiments that clearly explicate the roles of the sigma factors and RpaB in mediating circadian gene expression.

Although complex molecular mechanisms underlie the light-responsive expression of dusk genes, we demonstrate that phenomenological models effectively describe the differential activation of large groups of dusk genes to afternoon Shade and Sunset. These models suggest that transcription factors with the dynamics of RpaA~P and RpaB~P (*Figure 8A*) are sufficient to reproduce much of the activation of the Early, Middle, and Late clusters in response to a Shade pulse in the afternoon or Sunset just before night (*Figure 8C–E*). Our models suggest that additional feedback from the other gene clusters may be necessary to achieve the extent of circadian gating observed for the Early and Late clusters (*Figure 8G–I*). Our models suggest that interactions between the major dusk clusters can diversify the responses of these clusters to signals from RpaA and RpaB. Regulatory interactions between the sigma factors RpoD6, RpoD5, and SigF2 (*Figure 6—figure supplement 1*), which belong to the Early, Middle, and Late clusters, respectively, could generate feedback downstream of RpaA and RpaB similar to that in our models (*Figure 8—figure supplements 2–4*) to generate the diverse responses of the dusk clusters to light conditions. However, feedback could also come from other sources with similar dynamics to the cluster expression levels. Indeed we could not simultaneously fit our models to all four light conditions, likely because of global growth-rate-dependent differences between the Low Light and Clear Day conditions. Thus, complete modeling of transcription dynamics of light-dependent dusk genes likely requires explicitly including the effects of metabolism and growth on gene expression (*Reimers et al., 2017*; *Burnap, 2015*; *Scott et al., 2010*).

## Closing remarks

RpaB and its cognate upstream histidine kinase NblS (*van Waasbergen et al., 2002*) have been implicated in a variety of stress responses (*Marin et al., 2003*; *Mikami et al., 2002*; *Shoumskaya et al., 2005*), which suggests that the mechanisms and regulatory circuits defined here may apply to other environmental changes such as temperature or osmolarity. The requirement of RpaB for mediating the environmental response of circadian genes suggests that the circadian circuit coevolved with RpaB to optimize responses to predictable and unpredictable changes in the environment and motivates the further exploration of the interaction between light and circadian rhythms in *S. elongatus*. Resolution of this interaction and subsequent integration into whole cell models of cyanobacterial growth (*Burnap, 2015*; *Westermark and Steuer, 2016*) will help to explain the fitness benefits of the circadian clock (*Johnson and Egli, 2014*) and optimize synthetic biology efforts to engineer cyanobacteria to produce useful compounds (*Ducat et al., 2011*) from the constantly changing sunlight in nature.

## Genomics data

All high throughput sequencing data is available from the Gene Expression Omnibus with the accession number GSE104204.

# Materials and methods

The resources table includes the genetically modified organisms and strains, cell lines, reagents, and software that are essential to reproduce the results presented.

**Key resources table**

| Reagent type or resource | Designation | Source or reference | Identifiers | Additional information |
|---|---|---|---|---|
| Strain, strain background (*Synechococcus elongatus*) | PCC 7942 (wild-type) | ATCC | Cat. Num. 33912 | |
| Strain, strain background (*Escherichia coli*) | Tuner (DE3) | EMD Millipore | Cat. Num. 70263 | |
| Gene (*S. elongatus*) | RNA polymerase Beta' subunit | N/A | Cyanobase: *Synpcc7942_1524* | |
| Gene (*S. elongatus*) | *rpaB* | N/A | Cyanobase: *Synpcc7942_1453* | |
| Recombinant DNA reagent | RNA polymerase beta prime subunit FLAG | This paper | Addgene: 102337 | Plasmid encoding C-terminal FLAG tag RNA polymerase Beta' subunit (*Synpcc7942_1524*) with Kan selection marker, targeted to integrate at native gene locus |
| Recombinant DNA reagent | pET-48b(+) | EMD Millipore | Cat. Num. 71462 | |
| Renetic reagent (*S. elongatus*) | EOC 398 and EOC 399 | This paper | | *S. elongatus* PCC7942 transformed with RNA polymerase beta prime subunit FLAG plasmid. Confirmed by PCR and Western blot. |
| Antibody | anti-RpaB | This paper | | Anti-RpaB serum was produced by Cocalico Biologicals. Anti-RpaB was affinity purified as described in this work. |
| Antibody | anti-RpaA | This paper | | Anti-RpaA serum was produced by Cocalico Biologicals as described in *Markson et al., 2013*. Anti-RpaA was affinity purified as described in this work. |
| Antibody | FLAG M2 mouse monoclonal antibody | Sigma Aldrich | Cat. Num. F3165 | |

*Continued on next page*

*Continued*

| Reagent type or resource | Designation | Source or reference | Identifiers | Additional information |
|---|---|---|---|---|
| Software, algorithm | Imagequant | GE Healthcare | | |
| Software, algorithm | Bowtie | PMID: 19261174 | | |
| Software, algorithm | Peak-Seq | PMID: 19122651 | | |
| Software, algorithm | MATLAB | MathWorks | | |
| Commercial assay or kit | RNeasy Mini kit | Qiagen | Cat. Num. 74104 | |
| Commercial assay or kit | Ribo-Zero bacteria rRNA removal kit | Illumina | Cat. Num. MRZMB126 | |
| Commercial assay or kit | Truseq Stranded mRNA sample prep kit | Illumina | Cat. Num. 20020594 | |
| Commercial assay or kit | NEBNext Ultra II DNA library prep kit | New England Biolabs | Cat. Num. E7645S | |
| Chemical compound, drug | Phos-tag Acrylamide AAL-107 | Wako Pure Chemical Industries | Cat. Num. 304–93521 | |

## Cyanobacterial strains

Most experiments were conducted in a pure wildtype background of *Synechococcus elongatus* PCC7942 (ATCC catalog number 33912, RRID:SCR_001672). For RNAP ChIP experiments, we used a strain in which the $\beta'$ subunit of RNA polymerase (*Synpcc7942_1524*, gene info available through Cyanobase, RRID:SCR_007615) was C-terminally tagged with a 3x FLAG epitope (a gift from Ania Puszynska). To make this strain, wildtype *S. elongatus* was transformed with a plasmid encoding the *Synpcc7942_1524* gene with sequence encoding a 3X GS linker and a 3X FLAG epitope inserted before the stop codon, targeted to insert at the native locus of the gene. A downstream kanamycin resistance cassette was used for selection. This plasmid is available through Addgene with the ID 102337. Two different clones of this strain, EOC398 and EOC399, were confirmed by sequencing colony PCR fragments that amplified the modified regions of the gene, and the presence of the tagged subunit was confirmed with Western blotting.

## Construction of light apparatus

To grow the cyanobacteria in different light profiles, we constructed an apparatus to control the intensity of four high powered LED arrays (parts list in *Table 3*, p. 2). 'Warm white' LED arrays (~1 in. x 1 in., Bridgelux) were chosen because of maximal overlap with the phycobilisome absorption spectrum. An LED array was mounted on a heatsink (Nuventix) and powered by a Flexblock LED

**Table 3.** Parts for controllable light source.
The table includes the parts chosen for their specific properties. The remaining parts, such as wires, heat shrink tubing, thermal paste for mounting the LEDs on the heat sinks, proto-boards, and housing are quite general and specific brands are unnecessary.

| Part name | Digikey part number | Current price ($) | Quantity |
|---|---|---|---|
| PWR SUP MEDICAL 18V 8.3A 150W | EPS439-ND | 73.71 | 1 |
| CONN RCPT 8CONT DIN SLD PNL MNT | SC2007-ND | 5.64 | 1 |
| LEDDynamics Flexblock BUCK BOOST 48V, 700 mA | 788–1038-ND | 19.99 | 4 |
| AD7376 digital potentiometer | AD7376ARWZ10-ND | 8.66 | 4 |
| AC to DC power supply, 10VDC, 275 mA | 993–1233-ND | 4.68 | 2 |
| BXRA-30E1200-B-03, Bridgelux, Warm white, LED | Not sold at Digikey. | | |
| | Need to order from: | 10.47 | 4 |
| | AMBIT ELECTRONICS, INC. | | |
| Aavid thermalloy Spotlight 47W heat sink | 1061–1092-ND | 9.50 | 4 |
| Arduino Uno Board Rev3 | 1050–1024-ND | 21.49 | 1 |

DOI: https://doi.org/10.7554/eLife.32032.040

driver (LEDdynamics) wired in the 'boost only' configuration (*Table 4*, p. 3). The intensity of the LEDs was controlled by varying the voltage input into the DIM line of the Flexblock between 0 and 10 V. We used a digital potentiometer (AD7376, Analog Devices) as a controllable 10 V source. The voltage output of the digipot was controlled via serial peripheral interface with an Arduino Uno board (Arduino) (see *Table 5*, p. 4). Each LED array was controlled separately, and a single array was sufficient to grow a single 750 mL culture of *S. elongatus*. All wires carrying substantial currents from the main power supply to the LED arrays were rated 18 AWG, and all other wires were rated 22 AWG. The relatively low voltage of the main power supply (18 V) is essential for being able to turn off the LED arrays completely.

## Calibrating light conditions

A single LED was mounted to shine perpendicular to the ground and isolated from other light sources. A single 750 mL cyanobacterial culture in a 150 $cm^2$ BD Falcon Tissue culture flask (Fisher Scientific) was placed beneath the LED, tilted such that the broad face of the culture was almost perpendicular to the incoming light. Each LED was calibrated by passing a known voltage input to the LEDs and recording the intensity of the light in $\mu$mol photons $m^{-2}$ $s^{-1}$ at the position of the surface of the culture directly beneath the LED using a LI-COR LI-250A light meter equipped a quantum sensor. To access a greater dynamic range of light intensity values, we calibrated the lights to give light intensity values at either of two distances from the light source — raised towards the lights to access higher light intensities, or lowered away from the lights to access lower light intensities.

To define the Clear Day conditions, we used light intensity values measured by the Ground-based Atmospheric Monitoring Instrument Suite, Rooftop Instrument Group on March 23rd, 2013 (*Figure 1B*, dark blue line, [*Petty and Weidner, 2017*]). We used this light intensity profile to define the rate of change of light intensity in our Clear Day condition, with a maximal light intensity of 600 $\mu$mol photons $m^{-2}$ $s^{-1}$. This intensity is consistent with measurements of light intensity in aquatic environments (*Waldbauer et al., 2012*), while also offering an order of magnitude difference in intensity compared to the Low Light condition, which was a constant 50 $\mu$mol photons $m^{-2}$ $s^{-1}$. The Shade pulse condition was defined by dividing the intensity value of our Clear Day profile by 10 fold between 8 and 9 hr after dawn. The High Light pulse was defined as the intensity of the Clear Day condition between 8 and 9 hr after dawn. Low Light cultures were grown continuously at 50 $\mu$mol photons $m^{-2}$ $s^{-1}$. We generated the dynamic changes in light intensity of our conditions by changing the intensity of the LED every three minutes by passing the calibrated voltage value corresponding to the appropriate light intensity of our defined profile. The light intensity values of the Low Light and Clear Day conditions are listed in *Figure 2—source data 1*, and the High Light and Shade pulse values are listed in *Figure 3—source data 1*. After the 12 hr light profile, the LEDs were turned off for 12 hr during the dark period. Cultures were grown semi-turbidostatically ($OD_{750}$ maintained at 0.3) with periodic dilution in BG-11M media supplemented with 10 mM HEPES pH 8.0 at 30 °C, continuously bubbled with 1% $CO_2$ in air, and shaken at 25 rpm in an enclosure impermeable to room lighting. Cells were not grown with antibiotics during the course of the experiment.

**Table 4.** Wiring the FlexBlock LED driver.
The FlexBlock LED driver needs to be connected in a 'boost only' configuration (see spec sheet for more details), with connections as shown.

| Line | Connection |
| --- | --- |
| DIM GND | GND of 10 V power supply/Arduino |
| DIM | Wipe of AD7376 potentiometer (Pin 16) |
| Vin+ | +of 18V power supply AND + of LED array |
| Vin- | GND of 18V power supply |
| LED+ | NC (not connected) |
| LED- | - of LED array |

DOI: https://doi.org/10.7554/eLife.32032.041

**Table 5.** Wiring the AD7376 potentiometer.

We used the SOIC-16 housing for the AD7376 potentiometer for ease of soldering to wires. The table indicates how each pin was connected. The length of the GND wire from the Arduino board to the shared ground needs to be kept short (~2 in. or less) for SPI communication.

| Pin | Connection |
| --- | --- |
| 1 | +of 10 V power supply |
| 2 | GND (shared GND between that of 10V power supply and Arduino |
| 3 | GND |
| 4 | GND |
| 5 | pin 10 on Arduino (or any other pin designated as a Slave Select, such as 5, 6, or 9 |
| 6 | +5V of Arduino |
| 7 | pin 13 on Arduino (SCLK) |
| 8 | NC (not connected) |
| 9 | NC |
| 10 | NC |
| 11 | pin 11 on Arduino (MOSI) |
| 12 | +5V of Arduino |
| 13 | NC |
| 14 | +of 10V power supply |
| 15 | NC |
| 16 | DIM line of FlexBlock |

DOI: https://doi.org/10.7554/eLife.32032.042

## Purification of anti-RpaA and anti-RpaB antibodies

Recombinant RpaA was purified as previously described (*Takai et al., 2006*). To purify recombinant RpaB, we cloned the *rpaB* gene (*Synpcc7942_1453*, gene info available through Cyanobase, RRID: SCR_007615) into the pET48-b + plasmid (Novagen) and overexpressed Trx-His-tagged RpaB in Novagen Tuner (DE3) competent cells carrying this plasmid by adding 300 $\mu$M IPTG to mid-log phase cultures. RpaB was purified from cell lysate using Ni-NTA chromatography as described previously (*Gutu and O'Shea, 2013*). The Trx-His tag was cleaved from RpaB and removed using a subsequent Ni-NTA step as described (*Gutu and O'Shea, 2013*). Purified, cleaved RpaB was dialyzed into a buffer containing 20 mM HEPES-KOH, pH 8.0, 150 mM KCl, 10% w/v glycerol, and 1 mM DTT. Protein concentration was measured with the Pierce BCA assay, and aliquots were flash frozen and stored at −80°C .

Anti-RpaB serum was generated by immunization of two rabbits with purified RpaB by Cocalico Biologicals (Reamstown, PA). RpaA- and RpaB-conjugated Affigel 10/15 resin (Bio-Rad) was prepared following manufacturer's instructions as described previously (*Gutu and O'Shea, 2013*). Anti-RpaB serum was first passed over an RpaA-conjugated resin and the flowthrough collected to subtract cross-reacting antibodies. Anti-RpaB antibodies were then purified from the flowthrough using an RpaB-conjugated resin as described previously (*Gutu and O'Shea, 2013*). The same process was repeated to purify anti-RpaA antibodies using rabbit serum described previously (*Markson et al., 2013*), passing the serum over an RpaB-conjugated resin and purifying with an RpaA-conjugated resin. No cross reactivity of the purified anti-RpaA and anti-RpaB antibodies for the opposite regulator was detected via ELISA assay.

## Measurement of RpaA~P and RpaB~P levels

Ten mL of cyanobacterial culture with $OD_{750} = 0.3$ were collected on cellulose acetate filters and flash frozen prior to storage at −80 °C. Cell lysates for Western blotting were prepared from the collected cells as described previously (*Markson et al., 2013*). Equal amounts of cell lysate (10–15 $\mu$g) were resolved on Phos-tag acrylamide gels (Wako Laboratory Chemicals) and transferred to nitrocellulose membranes as described previously (*Gutu and O'Shea, 2013*). Membranes were probed with

1/5000 dilution of purified anti-RpaA and anti-RpaB antibody. RpaA blots were then incubated with goat anti-rabbit HRP-conjugated secondary antibody and developed using the Pierce Femto chemiluminescence kit. The exposed blots were imaged with an Alpha Innotech Imaging station. RpaB blots were incubated with Goat anti-Rabbit Westerndot 585 antibody (RRID:AB_2556786) and imaged with a Typhoon Imager. The intensities of the bands corresponding to unphosphorylated and phosphorylated RpaA/B were quantified using Imagequant software (GE Healthcare Life Sciences, RRID:SCR_014246) using rubber band background subtraction. The percent of RpaA (or RpaB) phosphorylated was quantified as the intensity of the RpaA~P band divided by the sum of the intensities of the RpaA and RpaA~P bands, multiplied by 100. Values reported in *Figures 4A,B and* and *5A,B* represent the average of two separate measurements from replicate Western blots, with error bars displaying the range of the measured values (See *Figure 4—source data 1*, and *Figure 5—source data 1* for raw data from the replicate experiments). The trends seen were reproducibly observed between separate biological replicates of the light condition time courses.

## RNA sequencing

Twenty-five mL of cyanobacterial culture with $OD_{750} = 0.3$ were collected on cellulose acetate filters and flash frozen prior to storage at $-80\ ^{\circ}C$. Cells were resuspended in RNAprotect Bacteria reagent (Qiagen), and 1/3 of the cells were resuspended in a buffer containing 15 mg/mL lysozyme, 10 mM Tris-Cl, 1 mM EDTA pH 8, and 50 mM NaCl and incubated for 10 min. RNA was purified from the lysed cells using the Qiagen RNeasy Mini Kit. Ribosomal RNA was depleted from 1.25 $\mu$g of purified RNA using the Ribo-Zero bacteria rRNA removal kit (Illumina). Strand-specific RNAseq libraries were prepared from the depleted RNA using the Truseq Stranded mRNA Sample prep kit (Illumina) and sequenced on an Illumina HiSeq 2500 machine by the Bauer Core Facility at the Harvard FAS Center for Systems Biology. Sequencing reads were aligned to the *S. elongatus* genome using Bowtie (RRID:SCR_005476) as described previously (*Markson et al., 2013*), with samples averaging 8 million aligned reads. We quantified expression of a gene by counting the number of aligned sequencing reads corresponding to the appropriate strand between the start and stop of each gene (gene info obtained from Cyanobase, RRID:SCR_007615), and normalized these values between all samples from the light conditions in this work using median normalization, followed by dividing the median normalized read count value by the length of the open reading frame of the gene, as described previously (*Anders and Huber, 2010*; *Markson et al., 2013*). The time course and RNA sequencing was repeated twice for two biological replicates (data available in *Figure 2—source data 1* and *Figure 3—source data 1*). The data plotted in this work are from replicate 2, and the trends observed are reproduced in both biological replicates.

## Definition of circadian genes

We defined a subset of previously identified circadian genes on which to focus our analysis. We began with a list of 856 previously described reproducibly circadian genes (*Markson et al., 2013*; *Vijayan et al., 2009*). We next required that these genes have a Cosiner amplitude (*Kucho et al., 2005*) of greater that 0.15 under Constant Light conditions (*Vijayan et al., 2009*). We also required that the gene display expression of at least one read per nucleotide in at least one time point of the RNA sequencing experiments in this study. These filters produce a list of 450 high confidence circadian genes.

We noted that genes classified as dawn (class 2) and dusk (class 1) genes under Constant Light conditions (*Vijayan et al., 2009*) showed maximal expression at a different time of day under our Low Light conditions, while the relative ordering of genes by Cosiner phase (*Kucho et al., 2005*) from Constant Light conditions (*Vijayan et al., 2009*) was preserved. As such, we redefined dawn genes as those genes with a phase of 40° to 189° under Constant Light conditions (*Vijayan et al., 2009*), and dusk genes as those with a phase of 190° to 360° and 0° to 39°, as determined by the Cosiner algorithm (*Kucho et al., 2005*). These definitions produce a list of 169 high confidence dawn genes, and 281 high confidence dusk genes. The expression of our redefined circadian genes under Constant Light conditions is plotted in *Figure 2—figure supplement 2*. The list of high confidence circadian genes and high confidence class assignments is available in *Figure 2—source data 1* and *Figure 3—source data 1*.

## ChIP sequencing

One hundred and twenty mL of OD$_{750}$ 0.3 cyanobacterial culture were removed and crosslinked with 1% formaldehyde at 30 °C for 5 min in front of a light source. Crosslinking was quenched with 125 mM glycine. Crosslinked cells were washed twice with phosphate buffered saline, pelleted, and flash frozen prior to storage at −80 °C.

Pellets were resuspended in 1 mL of BG-11M supplemented with 500 mM L-proline and 1 mg/mL lysozyme and incubated at 30 °C for 1 hr to digest the cell wall. Cells were collected and resuspended in a Lysis buffer (50 mM HEPES pH 7.5, 140 mM NaCl, 1 mM EDTA, 1% Triton X-100, 0.1% sodium deoxycholate, and 1x Roche Complete EDTA-free Protease Inhibitor Cocktail) prior to shearing in a Covaris E220 Adaptive Focus System (Peak Incident Power = 175; Duty Factor = 10%; Cycles per burst = 200; Time = 160 s). The lysates were cleared via centrifugation, and concentration was determined via the Pierce BCA Assay.

For a given pulldown, 800 $\mu$g of lysate was incubated overnight at 4 °C in 500 $\mu$L of lysis buffer with 8 $\mu$g of anti-RpaA, anti-RpaB, or FLAG M2 mouse monoclonal antibody (Sigma-Aldrich) for RNAP pulldowns. A mock pulldown was carried out in which equal amounts of lysate from every time point of the time course (Shade 0, 15, 60 min, High Light 0, 15, 60 min) in a total of 800 $\mu$g was incubated with 8 $\mu$g of rabbit Igg. Next, 35 $\mu$L of Dynabeads protein G (Thermo Fischer Scientific) equilibrated in lysis buffer were added and the sample was incubated with mixing for 2 hr at 4 °C. The beads were washed and DNA was eluted and purified as described previously (*Markson et al., 2013*).

Sequencing libraries were prepared from the purified ChIP DNA using the NEBNext Ultra II DNA Library Prep Kit (New England Biolabs, Ipswich, MA). Libraries were sequenced on an Illumina HiSeq 2500 instrument by the by the Bauer Core Facility at the Harvard FAS Center for Systems Biology. We created sequencing libraries of ChIP experiments from two separate biological repeats of the time course experiment. Reads were aligned to the *S. elongatus* genome using Bowtie (RRID:SCR_005476) as described previously (*Markson et al., 2013*), resulting in an average of 3 million aligned reads for replicate 1, and 5 million aligned reads for replicate 2.

## ChIP-seq analysis

The aligned read data per genomic position was smoothed with a Gaussian filter (window size = 400 base pairs, standard deviation = 50). Each data set was normalized to the Mock ChIP-seq experiment and peaks which were significantly enriched above the Mock were identified in each data set using a previously described (*Markson et al., 2013*) custom-coded form of the Peak-seq algorithm (*Rozowsky et al., 2009*). Within each replicate time course for a given protein, we compiled a list of peaks which were enriched at least 3.5 fold over the Mock experiment at the position of highest ChIP signal. Finally, we required that a peak be detected in both replicates for it to be considered. This analysis generated 114 RpaA peaks, 218 RpaB peaks, and 451 RNAP peaks. To calculate enrichment for a peak, we determined the ChIP signal at a given time point at the genomic position of the highest ChIP signal detected for that peak and divided this by the value of the Mock experiment at that position. The data plotted in this manuscript are from replicate 2, but all trends hold in replicate 1. We assigned a gene as a target of a peak if: (i) the start codon of the gene was within 500 bp of the position of maximal ChIP signal within a peak; (ii) the peak resided upstream of the gene; (iii) The gene was the closest gene to that peak on the same strand. Lists of RNAP, RpaA, and RpaB peaks and gene targets are found in *Figure 3—source data 2*, *Figure 4—source data 2*, and *Figure 5—source data 2*, respectively.

For *Figures 3G*, *4C* and *5C*, we identified all RNAP, RpaA, or RpaB peaks with dusk gene targets based on the above criteria, respectively. 82 dusk genes are targets of RNAP peaks, 56 dusk genes were targets of RpaA peaks, and 42 dusk genes are targets of RpaB peaks. Then, for each peak - dusk gene pair, we calculated the change in gene expression of the dusk gene after 60 min, and the change in ChIP enrichment of the upstream peak over the mock pulldown (described above) after 60 min in High light, each compared to their respective values at Low light at 8 hr since dawn. We plotted these data on the x- and y-axes, respectively, with orange triangles. We repeated this process, comparing gene expression and ChIP enrichment values after 60 min in Shade compared to 8 hr since dawn in Clear Day conditions, and plotted the data as gray circles. We calculated the correlation coefficient between the change in gene expression and the change in ChIP enrichment for all

peak-gene pairs of the relevant factor in the High Light pulse, and then calculated the same correlation in Shade pulse conditions separately. We calculated the correlation coefficients comparing changes after 15 min in either the High Light or Shade pulse conditions, and list these values in the legends of *Figure 3—figure supplement 2*, *Figure 4—figure supplement 2*, and *Figure 5—figure supplement 2*. The data used for these plots for RNAP, RpaA, and RpaB are available in *Figure 3—source data 2*, *Figure 4—source data 2*, and *Figure 5—source data 2*, respectively. We plot data from replicate 2, and the trends are reproduced in replicate 1.

For *Figure 3—figure supplement 2*, *Figure 4—figure supplement 2*, and *Figure 5—figure supplement 2*, we took the lists of RNAP/RpaA/RpaB peaks with dusk gene targets from above. For each peak - gene pair, we calculated the $\log_2$ fold change in ChIP enrichment of the peak and the change in expression of the downstream gene in 15 or 60 min in the High Light pulse compared to the value at 8 hr since dawn in Low Light conditions. We repeated these calculations for each peak-gene pair in 15 or 60 min in Shade pulse compared to 8 hr since dawn in Clear Day conditions. We used hierarchical clustering on the collective ChIP and gene expression data from both conditions to determine the plotting order of the peak-gene pairs in the heat maps, and then plotted the $\log_2$ change in ChIP enrichment and dusk target gene expression in the two conditions in separate heat maps. The change in enrichment of a peak and the change in expression of its target dusk gene are aligned horizontally in their respective heat maps. The leftmost column of each heat map is white, because this column compares the time 0 data to itself and thus has a $\log_2$ value of 0. One RpaA peak resides upstream of two dusk genes, and two RpaB peaks reside upstream of two dusk genes each, and thus the listed number of RpaA and RpaB peaks is smaller than the number of RpaA and RpaB target dusk genes. The data used for these plots for RNAP, RpaA, and RpaB are available in *Figure 3—source data 2*, *Figure 4—source data 2*, and *Figure 5—source data 2*, respectively. We plot data from replicate 2, and the trends are reproduced in replicate 1.

For *Figures 4D* and *5D* we identified all dusk genes that were targets of both RpaA and RNAP (for *Figure 4D*) or both RpaB and RNAP (for *Figure 5D*). 33 dusk genes are targets of both RpaA and RNAP peaks, and 27 dusk genes are targets of both RpaB and RNAP. Then, for each pair of RpaA/B - RNAP peaks, we calculated the change in ChIP enrichment of the RpaA/B peak after 60 min, and the change in ChIP enrichment of the RNAP peak upstream of the same dusk gene over the mock pulldown (described above) after 60 min in High light, each compared to their respective values at Low light at 8 hr since dawn. We plotted these data on the x- and y-axes, respectively, with orange triangles. We repeated this process, comparing RpaA/B ChIP enrichment and RNAP ChIP enrichment values after 60 min in Shade compared to 8 hr since dawn in Clear Day conditions, and plotted the data as gray circles. We calculated the correlation coefficient between the change in RpaA/B ChIP enrichment and the change in RNAP ChIP enrichment for all RpaA/B - RNAP peak pairs of the relevant factor in the High Light pulse, and then calculated the same correlation in Shade pulse conditions separately. We calculated the correlation coefficients comparing changes after 15 min in either the High Light or Shade pulse conditions, and list these values in the legends of *Figure 4—figure supplement 3*, and *Figure 5—figure supplement 3*. The RNAP, RpaA, and RpaB peaks associated with each dusk gene are listed in *Figure 2—source data 1* and *Figure 3—source data 1*, and the enrichment values for these peaks are listed in *Figure 3—source data 2*, *Figure 4—source data 2*, and *Figure 5—source data 2*, respectively. The data plotted here are from replicate 2, and the trends are reproduced in replicate 1.

For *Figure 4—figure supplement 3* and *Figure 5—figure supplement 3*, we took the lists of RpaA/RpaB - RNAP peaks pairs upstream of the same dusk gene from above. For each RpaA/B - RNAP peak, we calculated the $\log_2$ fold change in ChIP enrichment of the RpaA/B peak and the change in ChIP enrichment of the RNAP peak upstream of the same dusk gene in 15 or 60 min in the High Light pulse compared to the value at 8 hr since dawn in Low Light conditions. We repeated these calculations for each peak-gene pair in 15 or 60 min in Shade pulse compared to 8 hr since dawn in Clear Day conditions. We used hierarchical clustering on the collective RpaA/B and RNAP ChIP data from both conditions to determine the plotting order of the RpaA/RpaB - RNAP peak pairs in the heat maps, and then plotted the $\log_2$ change in RpaA/B ChIP enrichment and RNAP ChIP enrichment in the two conditions in separate heat maps. The change in enrichment of an RpaA/B peak and the change in enrichment of the RNAP peak upstream of the same dusk gene are aligned horizontally in their respective heat maps. The leftmost column of each heat map is white, because this column compares the time 0 data to itself and thus has a $\log_2$ value of 0. The RNAP,

RpaA, and RpaB peaks associated with each dusk gene are listed in *Figure 2—source data 1* and *Figure 3—source data 1*, and the enrichment values for these peaks are listed in *Figure 3—source data 2*, *Figure 4—source data 2*, and *Figure 5—source data 2*, respectively. The data plotted here are from replicate 2, and the trends are reproduced in replicate 1.

For *Figure 4—figure supplement 4D–F*, *Figure 6D*, and *Figure 6—figure supplement 1D–F*, we identified all RpaA, RpaB, and RNAP peaks that targeted the specified gene, as described above. Then, we calculated the $\log_2$ change in RpaA (dashed red line), RpaB (dotted blue line), RNAP (dashed green line) ChIP enrichment or expression of the downstream gene (solid black lines) in the High Light pulse compared to 8 hr since dawn in the Low Light condition, and plotted these values with downward triangles. We repeated these calculations, comparing enrichment and gene expression in the Shade pulse to the data at 8 hr since dawn in the Clear Day condition, and plotted these values with circles. The RNAP, RpaA, and RpaB peaks associated with each dusk gene are listed in *Figure 2—source data 1* and *Figure 3—source data 1*, and the enrichment values for these peaks are listed in *Figure 3—source data 2*, *Figure 4—source data 2*, and *Figure 5—source data 2*, respectively. The data plotted here are from replicate 2, and the trends are reproduced in replicate 1.

## K-means clustering

We calculated normalized expression values of high confidence dusk genes under our dynamic light conditions, as well as in previously described RpaA perturbations in Constant Light (*Markson et al., 2013*). We separately normalized the data from set of dynamic light conditions (Low Light, Clear Day, High Light pulse, Shade pulse) and the Constant Light data (Wildtype, OX-D53E cells — *rpaA*-, *kaiBC*-, *Ptrc::rpaA(D53E)* — without inducer, OX-D53E with inducer, [*Markson et al., 2013*]) using z-score normalization, and used this data to separate the dusk genes into eight groups with k-means clustering in MATLAB (RRID:SCR_001622) using Pearson correlation as the distance metric. We focused our analysis on the three largest clusters which accounted for most of the dusk genes (187/ 281 genes). The lists of genes belonging the three major clusters are found in *Figure 7—source data 1*.

## Mathematical modeling

We observed very regular and systematic changes in the expression of large clusters of dusk genes in natural light conditions (*Figures 2*, *3* and *7*) that correlated with $\mathrm{RpaA/B}$ recruitment of RNAP (*Figures 4–6*). Thus, our goal was to determine whether simple phenomenological models similar to that inspired by Alon (*Alon, 2006*) could reproduce these observations and offer some intuition into how they might arise. While most of the dusk genes underwent systematic changes, a small group of ~20 genes including *kaiBC* was relatively insensitive to changes in light intensity (*Figure 4—figure supplement 4*), and we do not model those genes' expression dynamics.

Our model treats the activation or repression of the expression of a dusk gene cluster by $\mathrm{RpaA{\sim}P}, \mathrm{RpaB{\sim}P}$, or another cluster using effective Hill kinetics. We coarse-grained each of the three groups of circadian dusk genes (the Early, Middle, and Late clusters in *Figure 7*) to a single effective gene with the average dynamics of the group (*Figure 7*, solid lines). We modeled the dynamics of a gene cluster $X$ using a simple kinetic model of an AND gate at a promoter (*Mangan and Alon, 2003*),

$$dX/dt = B_X + \beta_X f(\mathrm{RpaA{\sim}P}, K_{AX}, H_{AX}) f(\mathrm{RpaB{\sim}P}, K_{BX}, H_{BX}) f(Y, K_{YX}, H_{YX}) - \alpha_X X \qquad (1)$$

where $B_X$ is the basal transcription rate; $f$ is a function of the interaction of $X$ with $\mathrm{RpaA{\sim}P}, \mathrm{RpaA{\sim}P}$, or another cluster $Y$; $\beta_X$ is the max transcription rate; and $\alpha_X$ is the decay/dilution rate. Activating interactions were treated using a simple Hill function,

$$f(u, K, H) = (u/K)^H / (1 + (u/K)^H), \qquad (2)$$

where $u$ is the concentration of the active transcription factor, $H$ is the Hill coefficient of interaction, and $K$ is the coefficient of activation. Bacteria can easily tune the interactions between proteins and between transcription factors and promoters to adjust $H$ and $K$ for different clusters (*Buchler et al., 2003*). $\mathrm{RpaA{\sim}P}$ and $\mathrm{RpaB{\sim}P}$, were treated as activators, consistent with the results from *Figures 4– 6*. Repressive interactions between clusters were treated using

$$f(u,K,H) = 1/(1 + (u/K)^H), \tag{3}$$

where $K$ is now the coefficient of repression. In *Equation 1*, $\mathrm{RpaA{\sim}P}$, $\mathrm{RpaB{\sim}P}$, and $Y$ were measured experimentally; the remainder of the parameters were left free.

We determined the sufficiency of a model to describe the data by fitting the parameters using the range of values shown in *Table 1*. Time propagation of the differential *Equation 1* was performed using the *ode45* solver in MATLAB (RRID:SCR_001622), with $X(t=0)$ set as the observed expression level at the beginning of the simulated time period. Model fitting was performed in MATLAB using the non-linear least squares solver *lsqnonlin*.

The Akaike Information Criterion (AIC) and the Chi-squared test are typically used to quantify whether a model with more parameters fits the data better than another with fewer parameters simply because it is more complex. However, both approaches are for statistical models in which little to no information is used to construct the model and are not strictly applicable to the model constructed here, which is based on our understanding of transcription. If we do use AIC to compare the models, the feedback models are predicted to be most probable.

In our model, $H$ and $K$ are effective constants that represent the overall ability of $\mathrm{RpaA{\sim}P}$, $\mathrm{RpaB{\sim}P}$, or another gene cluster $Y$ to affect gene expression. These constants include potential indirect activation through the sigma factors, which is may be why joint activation by $\mathrm{RpaA{\sim}P}$ and $\mathrm{RpaB{\sim}P}$ describe the dynamics of the Middle cluster reasonably well. However, circadian gating of the Early and Late dusk genes requires further interactions that cannot be described by Hill functions of measured $\mathrm{RpaA{\sim}P}$ and $\mathrm{RpaB{\sim}P}$ levels. Clearly there may be more complex networks at play than those we have considered here, and much more needs to be done to fully model gene expression in *S. elongatus*. Here we have constructed a first model to suggest simple principles underlying the interaction of circadian and light regulation of dusk genes and offer directions for further exploration.

## Acknowledgements

JRP and KA thank Phil Shiu, Bin He, Eddie Wang, Andrian Gutu, Chris Chidley, Luca Gerosa, Vadim Patsalo, Rohan Balakrishnan, Matteo Mori, and Doran Bennett for helpful comments and discussions on the manuscript. We thank Andrew Kennard for initial experiments on the role of RpaB in regulating gene expression under dynamic light conditions. JRP thanks Christian Daly, Claire Reardon, Jennifer Couget, and Patrick Dennett of the Harvard Bauer Core Facility for their help with high throughput sequencing and other experiments. KA thanks Al Takeda and Jim MacArthur from the Harvard Electronics Shop for their extensive help in building the controllable lights. JRP and KA were supported by the Howard Hughes Medical Institute through EKO.

## Additional information

### Competing interests

Erin K O'Shea: President of Howard Hughes Medical Institute, one of the three founding funders of eLife. The other authors declare that no competing interests exist.

### Funding

| Funder | Author |
| --- | --- |
| Howard Hughes Medical Institute | Erin K O'Shea |

The funders had no role in study design, data collection and interpretation, or the decision to submit the work for publication.

### Author contributions

Joseph Robert Piechura, Conceptualization, Resources, Data curation, Formal analysis, Validation, Investigation, Visualization, Methodology, Writing—original draft, Writing—review and editing; Kapil

Amarnath, Conceptualization, Resources, Software, Formal analysis, Visualization, Methodology, Writing—original draft, Writing—review and editing; Erin K O'Shea, Conceptualization, Supervision, Funding acquisition, Project administration, Writing—review and editing

### Author ORCIDs
Joseph Robert Piechura  http://orcid.org/0000-0001-5349-4567
Kapil Amarnath  http://orcid.org/0000-0003-2589-9684
Erin K O'Shea  http://orcid.org/0000-0002-2649-1018

### Decision letter and Author response
Decision letter https://doi.org/10.7554/eLife.32032.051
Author response https://doi.org/10.7554/eLife.32032.052

## Additional files

### Supplementary files
• Transparent reporting form
DOI: https://doi.org/10.7554/eLife.32032.043

### Major datasets
The following dataset was generated:

| Author(s) | Year | Dataset title | Dataset URL | Database, license, and accessibility information |
|---|---|---|---|---|
| JR Piechura, K Amarnath, EK O'Shea | 2017 | Natural changes in light interact with circadian regulation at promoters to control gene expression in cyanobacteria | https://www.ncbi.nlm.nih.gov/geo/query/acc.cgi?acc=GSE104204 | Publicly available at the NCBI Gene Expression Omnibus (accession no: GSE104204) |

The following previously published datasets were used:

| Author(s) | Year | Dataset title | Dataset URL | Database, license, and accessibility information |
|---|---|---|---|---|
| V Vijayan, R Zuzow, EK O'Shea | 2009 | Oscillations in supercoiling drive circadian gene expression in cyanobacteria | https://www.ncbi.nlm.nih.gov/geo/query/acc.cgi?acc=GSE18902 | Publicly available at the NCBI Gene Expression Omnibus (accession no: GSE18902) |
| JS Markson, JR Piechura, AM Puszynska, EK O'Shea | 2013 | Circadian control of global gene expression by the cyanobacterial master regulator RpaA | https://www.ncbi.nlm.nih.gov/geo/query/acc.cgi?acc=GSE50922 | Publicly available at the NCBI Gene Expression Omnibus (accession no: GSE50922) |

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
