## [Decision Letter]

Thank you for submitting your article "Natural changes in light interact with circadian regulation at promoters to control gene expression in cyanobacteria" for consideration by *eLife*. Your article has been reviewed by two peer reviewers, and the evaluation has been overseen by Naama Barkai as the Senior and Reviewing Editor. The following individuals involved in review of your submission have agreed to reveal their identity: David F Savage (Reviewer #1).

The reviewers have discussed the reviews with one another and the Reviewing Editor has drafted this decision to help you prepare a revised submission.

Since all comments are related to reformulation of the text, we kindly ask you to account for all suggestions.

*Reviewer #1:*

Cyanobacteria are notable for their circadian rhythm. Here, Piechura et al. investigate circadian rhythm using growth conditions that are more natural than typical protocols to reveal insights into the control and regulation of gene expression. The work is a significant advance and I am supportive of publication but have a number of concerns and questions as detailed below:

1) I applaud the authors for investigating this pathway in the context of a more natural lighting protocol but this does raise one concern regarding their strain. It is well appreciated (e.g. in their cited reference Yu et al., 2015 Sci. Reports) that the strain PCC7942 is poorly suited for growth at high light. Thus, the conditions the authors have chosen may also be arbitrary. It would be helpful to address this concern with additional (brief) background on why these conditions were deemed suitable in the context of PCC7942.

2) The conclusions are generally supported by the data throughout, but I did have a concern about the statement: '…mRNA levels primarily by regulating…transcription'. This has not been proven and the other possibility – changes to degradation rates (or dilution) – was not tested. The dramatic changes observed from the data (i.e. Figure 3) in just 15 min actually suggest to me the opposite. Namely, a rapid decrease on such short timescales can only be explained by degradation. Please clarify.

3) Beginning of Part C. It might clarify this rationale to highlight the data in Figure 2. The incongruence between this and levels of RpaA-P is a simpler justification for the narrative.

4) Figure 6 highlights the fact that only a minor number of dusk responsive genes (~25%) are directly regulated by RpaA / RpAB. The presented model at the end of the Discussion (and the model itself) invokes a reasonable solution by placing σ factors downstream of these master regulators. This certainly could be true, but I think it is worth considering there could be transient regulation that exists outside (or at least is semi-autonomous) of the circadian framework. E.g. there is already data on N availability (Herrero et al., 2001 J. Bact.) and light availability modulating the stringent response (Hood et al., 2016 PNAS) which should be considered.

*Reviewer #2:*

Evaluation of paper

The question this paper asks is of basic interest and it is good it is tackled head on. The introduction very clearly puts previous studies on the circadian rhythm in the context of the question this study approaches. The narrative of the Results and Discussion is full of details and I found myself easily lost, as I am not someone who studied the genes referred to in the past and thus find it hard to deal with all the terminology of specific genes. It is not easy to solve and not unique to this paper but maybe the authors could help the reader further on this issue. My detailed comments are below.

- There are many claims such as: "Levels of RpaA⇠P increased from dawn to dusk". Why are quantitative values not reported?

- The analysis refers to gene expression which I gather is the RNA level. It is highly informative to know what happens at the proteome level. Can the author give some indication of that? It seems like any proteomics analysis can complement the picture in an essential manner. Specifically, how deep is the modulation of protein levels across the circadian rhythm?

- There are quite a few figures in the paper, but I am missing seeing some raw expression levels plots (before delving into the supplementary information, which many readers will not do). I would hope to see say 5 genes with their temporal dynamics before all the grouping to clusters etc. This serves me as a reader in seeing what the data looks like and gives me intuition into the strength of modulation before the statistical manipulations.

---

## [Author Response]

Reviewer #1:Cyanobacteria are notable for their circadian rhythm. Here, Piechura et al. investigate circadian rhythm using growth conditions that are more natural than typical protocols to reveal insights into the control and regulation of gene expression. The work is a significant advance and I am supportive of publication but have a number of concerns and questions as detailed below:1) I applaud the authors for investigating this pathway in the context of a more natural lighting protocol but this does raise one concern regarding their strain. It is well appreciated (e.g. in their cited reference Yu et al., 2015 Sci. Reports) that the strain PCC7942 is poorly suited for growth at high light. Thus, the conditions the authors have chosen may also be arbitrary. It would be helpful to address this concern with additional (brief) background on why these conditions were deemed suitable in the context of PCC7942.

We thank the reviewer for this comment and agree that the light conditions we chose for this study are somewhat arbitrary due to the lack of information about the natural ecological niche of *Synechococcus elongatus* PCC7942. However, we can argue that both the changes and the dynamic range of intensities we used are relevant for *S. elongatus*. First, we argue that the gradual changes in intensity due to the earth’s rotation in our Clear Day condition, and the rapid changes in light in our Shade pulse and High Light pulse conditions due to changes in cloud cover are relevant for any organism exposed to ambient light in nature, as shown by the measured sunlight dynamics in Figure 1. Second, we measured the growth rate of *Synechococcus elongatus* PCC7942 under both of our Low Light and Clear Day conditions after two days of acclimation to the respective condition (Figure 2—figure supplement 1) and found that the strain grows roughly twice as fast under the higher light intensities of the Clear Day condition at 6 hours after dawn compared to growth under the Low Light condition. These data indicate that *Synechococcus elongatus* PCC7942 is capable of acclimating to the higher light intensities of our Clear Day condition and using this energy for faster growth compared to lower light intensities. Thus, we added a sentence to the main text in the first paragraph of section A of the Results to highlight this comparison – ‘Cultures grown under the Clear Day condition adjusted their pigment content after two days of exposure to the Clear Day condition (Figure 2—figure supplement 1). […] These data indicate that *S. elongatus* PCC7942 is capable of acclimating to the higher light intensities of the Clear Day condition and thus that the intensities used in our measurements are relevant for this strain.’

2) The conclusions are generally supported by the data throughout, but I did have a concern about the statement: '…mRNA levels primarily by regulating…transcription'. This has not been proven and the other possibility – changes to degradation rates (or dilution) – was not tested. The dramatic changes observed from the data (i.e. Figure 3) in just 15 min actually suggest to me the opposite. Namely, a rapid decrease on such short timescales can only be explained by degradation. Please clarify.

We thank the reviewer for this comment and agree that mRNA degradation or dilution may play a part in generating the rapid changes in mRNA levels we observe in response to pulse changes in light. We observe that both RNA polymerase and RpaA/B occupancy at promoters tracks closely with these fast dynamics (Figure 3, Figure 3—figure supplement 2, Figure 4, Figure 4—figure supplement 2 and Figure 4—figure supplement 3, Figure 5, and Figure 5—figure supplement 2 and Figure 5—figure supplement 3). Moreover, genome-wide measurements of mRNA half lives in bacteria such as *E. coli* (Chen et al., 2015, DOI 10.15252/msb.20145794) and *B. subtilis* (Hambraeus et al., 2003, DOI: 10.1007/s00438-003-0883-6) reveal that most transcripts have half lives on the order of 3-7 minutes and are thus highly unstable, and measurements of single transcript half lives in *S. elongatus* PCC7942 fall within this range (Salem et al., 2004, DOI: 10.1128/JB. 186.6.1729-1736.2004). This rate of degradation is an order of magnitude greater than the growth rate of *S. elongatus*, and thus it dominates over loss of transcripts due to dilution. Taken together, these data strongly suggest a model in which there is a high constant degradation rate of transcripts, with the rate of synthesis tightly controlled in response to circadian and light-responsive regulation. Restated more specifically in the context of the example you cite, our model is that changes in the regulation of RNA Polymerase recruitment to dusk genes after the exposure to High Light leads to a rapid decrease in the synthesis rate of dusk mRNAs, and basal rapid turnover of dusk gene mRNAs causes a rapid decrease in mRNA levels. However, we have not directly tested genome-wide mRNA half lives under our conditions, and cannot rule out that some transcripts have regulated changes in degradation under our conditions. We clarify this model by adding to the text in the last paragraph of section B of the Results – ‘To cause these reversible changes in the mRNA levels of dusk genes, changes in light intensity must affect either the transcription *and/*or the degradation of dusk gene mRNAs. […] Because mRNAs in bacteria have very short steady state half lives (Chen et al., 2015, Hambraeus et al., 2003, Salem et al., 2004), we argue that changes in transcription rates of dusk gene mRNAs are sufficient to lead to the rapid changes in dusk gene mRNA levels given a fast basal degradation rate, though we cannot rule out that changes in light may affect the rates of degradation of some mRNAs.’

3) Beginning of Part C. It might clarify this rationale to highlight the data in Figure 2. The incongruence between this and levels of RpaA-P is a simpler justification for the narrative.

We thank the reviewer for this suggestion and agree that this argument helps motivate Part C. Thus, we now begin the first paragraph of section C of the Results as follows – ‘Given the strong dependence of dusk gene expression on RpaA~P levels under Constant Light conditions (Figure 1, Markson et al., 2013) and the drastic change in dusk gene expression dynamics under our dynamic light conditions (Figure 2; Figure 3), we hypothesized that light conditions alter RpaA~P dynamics to alter dusk gene expression. However, levels of RpaA~P increased from dawn to dusk... by the Kai PTO (Figure 4). […] Interestingly, ChIP-seq showed that light intensity fluctuations alter RpaA~P binding upstream of dusk genes (Figure 4; Figure 4—figure supplement 2) in conjunction with RNAP binding upstream of the same gene (Figure 4; Figure 4—figure supplement 3).’ We also found a couple of other places in the text where we could make the larger narrative a bit more clear.

4) Figure 6 highlights the fact that only a minor number of dusk responsive genes (~25%) are directly regulated by RpaA / RpAB. The presented model at the end of the Discussion (and the model itself) invokes a reasonable solution by placing σ factors downstream of these master regulators. This certainly could be true, but I think it is worth considering there could be transient regulation that exists outside (or at least is semi-autonomous) of the circadian framework. E.g. there is already data on N availability (Herrero et al., 2001 J. Bact.) and light availability modulating the stringent response (Hood et al., 2016 PNAS) which should be considered.

We thank the reviewer for this comment and agree that it is entirely possible that dusk gene expression is affected by transcriptional regulatory pathways that are independent of circadian regulation downstream of RpaA. The reviewer’s reference for light affecting the stringent response is particularly salient for our work. We altered the last paragraph of Section C of the Results to read as follows: ‘It is also possible that changes in light intensity affect dusk gene expressionindependently of RpaA~P and RpaB~P. For instance, global growth-rate-dependent gene regulatory mechanisms such as the stringent response (Scott et al., 2010; Ryals et al., 1982, Hood et al., 2016, Burnap et al., 2015), likely cause some of the light-dependent changes in circadian gene expression…” As shown in Ryals et al., 1982 and Scott et al., 2010, the current understanding of the stringent response in *E. coli* is that ppGpp levels respond to the total metabolic flux of the cell, the growth rate. Burnap et al., 2015 extends this understanding to cyanobacteria, and Hood et al., 2016 shows that the effects of changes in ppGpp levels in cyanobacteria resembles those observed for *E. coli*.

Reviewer #2:Evaluation of paperThe question this paper asks is of basic interest and it is good it is tackled head on. The introduction very clearly puts previous studies on the circadian rhythm in the context of the question this study approaches. The narrative of the Results and Discussion is full of details and I found myself easily lost, as I am not someone who studied the genes referred to in the past and thus find it hard to deal with all the terminology of specific genes. It is not easy to solve and not unique to this paper but maybe the authors could help the reader further on this issue. My detailed comments are below.- There are many claims such as: "Levels of RpaA⇠P increased from dawn to dusk". Why are quantitative values not reported?

We thank the reviewer for this comment. We added (or adjusted) quantified metrics to the text in the following places:

Second paragraph of Section A of the Results – “However, in the Clear Day condition 159of the 281 dusk genes were expressed two fold or higher after middaycompared to Low Light, demonstrating light-de- pendent expression.”

Last paragraph of section A of the Results – “Remarkably, though in both light conditions the cells experience 50 µmol photons m−2 s −1 at the end of the day just before night, light-dependent dusk genes have substantially higher expression in the Clear Day conditions relative to the Low Light conditions (Figure 2). Indeed, 95/281 dusk genes were expressed at least 3 fold higher in Clear Day relative to Low Light at 12 hours after dawn.”

First paragraph of section B of the Results – “The expression of dusk genes rapidly changed in a direction opposite to the change in light intensity (Figure 3, all dusk genes; Figure 3, example dusk gene; Figure 3, all dusk genes; Figure 3, example dusk gene), as expected from the effects of the decrease in light intensity at Sunset of the Clear Day condition on circadian gene expression (Figure 2). […] Further, many genes responded rapidly and changed in expression at least 3 fold after just 15 minutes into the pulse (75/281 repressed by High Light, 79/281 induced by Shade). When cultures were restored to their original condition...”

Paragraph 3 of section C of the Results – “We observed that levels of RpaB∼P changed rapidly in a direction opposite to the change in light (Figure 5; Figure 5—figure supplement 1), suggesting that light affects RpaB activity through its phosphorylation state (Figure 5). […] This strong correlation between RpaB~P levels and the expression of dusk genes under dynamic light conditions (also compare Figure 3 to Figure 5) suggests that RpaB~P acts as an activator of dusk gene expression. Indeed, using ChIP-seq, we found that...”

Since RpaA~P levels do not substantially change in the light conditions we use in this study, we opted against including quantified comparisons in the text. We have made available the measured values of RpaA and RpaB phosphorylation and all quantitative data of gene expression and RpaA, RpaB, and RNA Polymerase ChIP enrichment under all measured conditions for 2 replicate experiments in supplementary data files.

- The analysis refers to gene expression which I gather is the RNA level. It is highly informative to know what happens at the proteome level. Can the author give some indication of that? It seems like any proteomics analysis can complement the picture in an essential manner. Specifically, how deep is the modulation of protein levels across the circadian rhythm?

We thank the reviewer for this comment and agree that a full understanding of how gene regulation contributes to changes in physiology in cyanobacteria must incorporate information of how and the extent to which mRNA levels affect the abundance of the relevant proteins. However, at present, there is little known about this relationship in *S. elongatus* PCC7942. Proteins such as the σ factor RpoD3 have been shown to increase in abundance after just 30 minutes of high light exposure in conjunction with induction of the *rpoD3* mRNA (Seki, et al., 2007, DOI: 10.1074/jbc.M707582200), suggesting that rapid changes in mRNA levels can lead to changes in protein levels in cyanobacteria. Still, there is limited proteomic data (Guerrero, Mol Cell Proteomics, 2015) on *S. elongatus* both in constant light and variable light conditions. Hence, we agree and suggest that careful proteomic work is a critical next step in exploring the physiology of cyanobacteria under natural conditions. As these experiments are beyond the scope of our current study, we suggest this work for future study in the Discussion.

- There are quite a few figures in the paper, but I am missing seeing some raw expression levels plots (before delving into the supplementary information, which many readers will not do). I would hope to see say 5 genes with their temporal dynamics before all the grouping to clusters etc. This serves me as a reader in seeing what the data looks like and gives me intuition into the strength of modulation before the statistical manipulations.

We thank the reviewer for this comment and are committed to making the data transparent to readers so that they can fully comprehend the magnitude of expression changes to the several hundred dusk genes we analyze in this study. Here we point out our efforts to do so graphically in addition to the quantified metrics we added to the text from the previous comment. In an effort to impress upon the reader the magnitude of the expression of changes to this large group of genes, we highlight the expression of a single example gene in Figure 2 and Figure 3, and then present the expression of all 281 dusk genes in heat maps in which the data are expressed as the log change in expression from the average expression of the gene under the Low Light condition. This normalization allows a direct comparison of the magnitude of change of the dusk genes from a standard expression level that is specific to each gene. To allow a direct comparison between the example dusk gene for which we show raw data in Figure 2 and Figure 3 (*Synpcc7942_1567*), we indicate the position of this gene in the heat maps with an arrow. This arrow allows the reader to identify large groups of genes within the heatmaps that show comparable magnitude changes to the example dusk gene *Synpcc7942_1567.* Further, we use scatter plots to demonstrate the magnitude of change in expression of circadian genes after exposure to the High Light and Shade pulses in Figure 3, Figure 4, Figure 5 (subsets of genes), and 6B (all dusk genes). Figure 6 shows the full range of responses of dusk genes to rapid changes in light intensity, and demonstrates that many dusk genes change in expression by 2 fold or greater after exposure to the High Light and Shade Pulse, but some genes are not as drastically affected. Further, we show raw expression data for 8 other dusk genes in Figure 2—figure supplement 4; Figure 4—figure supplement 4; and Figure 6—figure supplement 1. Finally, all raw gene expression data for all of our conditions for two different biological replicate experiments are available for download in supplemental data files.